# CHARACTERIZING THE INFLUENCE OF GRAPH ELEMENTS

**Zizhang Chen, Peizhao Li, Hongfu Liu, Pengyu Hong**
Brandeis University
{zizhang2,peizhaoli,hongfuliu,hongpeng}@brandeis.edu

## ABSTRACT

Influence function, a method from robust statistics, measures the changes of model parameters or some functions about model parameters concerning the removal or modification of training instances. It is an efficient and useful post-hoc method for studying the interpretability of machine learning models without the need for expensive model re-training. Recently, graph convolution networks (GCNs), which operate on graph data, have attracted a great deal of attention. However, there is no preceding research on the influence functions of GCNs to shed light on the effects of removing training nodes/edges from an input graph. Since the nodes/edges in a graph are interdependent in GCNs, it is challenging to derive influence functions for GCNs. To fill this gap, we started with the simple graph convolution (SGC) model that operates on an attributed graph and formulated an influence function to approximate the changes of model parameters when a node or an edge is removed from an attributed graph. Moreover, we theoretically analyzed the error bound of the estimated influence of removing an edge. We experimentally validated the accuracy and effectiveness of our influence estimation function. In addition, we showed that the influence function of a SGC model could be used to estimate the impact of removing training nodes/edges on the test performance of the SGC without re-training the model. Finally, we demonstrated how to use influence functions to guide the adversarial attacks on GCNs effectively.

## 1 INTRODUCTION

Graph data is pervasive in real-world applications, such as, online recommendations (Shalaby et al., 2017; Huang et al., 2021; Li et al., 2021), drug discovery (Takigawa & Mamitsuka, 2013; Li et al., 2017), and knowledge management (Rizun, 2019; Wang et al., 2018), to name a few. The growing need to analyze huge amounts of graph data has inspired work that combines Graph Neural Networks with deep learning (Gori et al., 2005; Scarselli et al., 2005; Li et al., 2016; Hamilton et al., 2017; Xu et al., 2019b; Jiang et al., 2019). Graph Convolutional Networks (GCNs) (Kipf & Welling, 2017; Zhang & Chen, 2018; Fan et al., 2019), the most cited GNN architecture, adopts convolution and message-passing mechanisms.

To better understand GCNs from a data-centric perspective, we consider the following question:

> *Without model retraining, how can we estimate the changes of parameters in GCNs*
> *when the graph used for learning is perturbed by edge- or node-removals?*

This question proposes to estimate counterfactual effects on the parameters of a well-trained model when there is a manipulation in the basic elements in a graph, where the ground truth of such an effect should be obtained from model retraining. With a computational tool as the answer, we can efficiently manipulate edges or nodes in a graph to control the change of model parameters of trained GCNs. The solution would provide further extensions like graph data rectification, improving model generalization, and graph data poison attacks through a pure data modeling way. Yet, current methods for training GCNs offer limited interpretability of the interactions between the training graph and the GCN model. More specifically, we fall short of understanding the influence of the input graph elements on both the changes in model parameters and the generalizability of a trained model (Ying et al., 2019; Huang et al., 2022; Yuan et al., 2021; Xu et al., 2019a; Zheng et al., 2021).

In the regime of robust statistics, an analyzing tool called influence functions (Hampel, 1974; Koh & Liang, 2017) is proposed to study the counterfactual effect between training data and model performance. For independent and identically distributed (i.i.d.) data, influence functions offer an approximate estimation of the model's change when there is an infinitesimal perturbation added to the training distribution, *e.g.*, a reweighing on some training instances. However, unlike i.i.d. data, manipulation on a graph would incur a knock-on effect through GCNs. For example, an edge removal will break down all message passing that is supposed to pass through this edge and consequentially change node representations and affect the final model optimization. Therefore, introducing influence functions to graph data and GCNs is non-trivial work and requires extra considerations.

In this work, we aim to derive influence functions for GCNs. As the first attempt in this direction, we focused on Simple Graph Convolution (Wu et al., 2019). Our contributions are three-fold:

- We derived influence functions for Simple Graph Convolution. Based on influence functions, we developed computational approaches to estimate the changes in model parameters caused by two basic perturbations: edge removal and node removal.

- We derived the theoretical error bounds to characterize the gap between the estimated changes and the actual changes in model parameters in terms of both edge and node removal.

- We show that our influence analysis on the graph can be utilized to (1) rectify the training graph to improve model testing performance, and (2) guide adversarial attacks to SGC or conduct grey-box attacks on GCNs via a surrogate SGC.

Code is publicly available at `https://github.com/Cyrus9721/Characterizing_ graph_influence`.

## 2 PRELIMINARIES

In the following sections, we use a lowercase $x$ for a scalar or an entity, an uppercase $X$ for a constant or a set, a bolder lowercase $\mathbf{x}$ for a vector, and a bolder uppercase $\mathbf{X}$ for a matrix.

**Influence Functions** Influence functions (Hampel, 1974) estimate the change in model parameters when the empirical weight distribution of i.i.d. training samples is perturbed infinitesimally. Such estimations are computationally efficient compared to learn-one-out retraining iterating every training sample. For $N$ training instances $\mathbf{x}$ and label $y$, consider empirical risk minimization (ERM) $\hat{\theta} = \arg\min_{\theta \in \Theta} \frac{1}{N} \sum_{\mathbf{x},y} \ell(\mathbf{x}, y) + \frac{\lambda}{2} \|\theta\|_2^2$ for some loss function $\ell(\cdot, \cdot)$ through a parameterized model $\theta$ and with a regularization term. When down weighing a training sample $(\mathbf{x}_i, y_i)$ by an infinitely small fraction $\epsilon$, the substitutional ERM can be expressed as $\hat{\theta}(\mathbf{x}_i; -\epsilon) = \arg\min_{\theta \in \Theta} \frac{1}{N} \sum_{\mathbf{x},y} \ell(\mathbf{x}, y) - \epsilon \ell(\mathbf{x}_i, y_i) + \frac{\lambda}{2} \|\theta\|_2^2$. Influence functions estimate the actual change $\mathcal{I}^*(\mathbf{x}_i; -\epsilon) = \hat{\theta}(\mathbf{x}_i; -\epsilon) - \hat{\theta}$ for a strictly convex and twice differentiable $\ell(\cdot, \cdot)$:

$$\mathcal{I}(\mathbf{x}_i; -\epsilon) = \lim_{\epsilon \to 0} \hat{\theta}(\mathbf{x}_i; -\epsilon) - \hat{\theta} = -\mathbf{H}_{\hat{\theta}}^{-1} \nabla_{\hat{\theta}} \ell(\mathbf{x}_i, y_i), \tag{1}$$

where $\mathbf{H}_{\hat{\theta}} := \frac{1}{N} \sum_{i=1}^{N} \nabla_{\hat{\theta}}^2 \ell(\mathbf{x}_i, y_i) + \lambda \mathbf{I}$ is the Hessian matrix with regularization at parameter $\hat{\theta}$. For some differentiable model evaluation function $f : \Theta \to \mathbb{R}$ like calculating total model loss over a test set, the change from down weighing $\epsilon \to (\mathbf{x}_i, y_i)$ to the evaluative results can be approximated by $\nabla_{\hat{\theta}} f(\hat{\theta}) \mathbf{H}_{\hat{\theta}}^{-1} \nabla_{\hat{\theta}} \ell(\mathbf{x}_i, y_i)$. When $N$ the size of the training data is large, by setting $\epsilon = \frac{1}{N}$, we can approximate the change of $\hat{\theta}$ incurred by removing an entire training sample $\mathcal{I}(\mathbf{x}_i; -\frac{1}{N}) = \mathcal{I}(-\mathbf{x}_i)$ via linear extrapolations $\frac{1}{N} \to 0$. Obviously, in terms of the estimated influence $\mathcal{I}$, removing a training sample has the opposite value of adding the same training sample $\mathcal{I}(-\mathbf{x}_i) = -\mathcal{I}(+\mathbf{x}_i)$. In our work, we shall assume an additivity of influence functions in computations when several samples are removed, *e.g.*, when removing two samples: $\mathcal{I}(-\mathbf{x}_i, -\mathbf{x}_j) = \mathcal{I}(-\mathbf{x}_i) + \mathcal{I}(-\mathbf{x}_j)$.

Though efficient, as a drawback, influence functions on non-convex models suffer from estimation errors due to the variant local minima and usually a computational approximation to $\mathbf{H}_{\hat{\theta}}^{-1}$ for a non-invertible Hessian matrix. To introduce influence functions from i.i.d. data to graphs and precisely characterize the influence of graph elements to model parameters' changes, we consider a convex model called Simple Graph Convolution from the GCNs family.

**Simple Graph Convolution**   By removing non-linear activations between layers from typical Graph Convolutional Networks, Simple Graph Convolution (SGC) (Wu et al., 2019) formulates a linear simplification of GCNs with competitive performance on various tasks (He et al., 2020; Rakhimberdina & Murata, 2019). Let $G = (V, E)$ denote an undirected attributed graph, where $V = \{v\}$ contains vertices with corresponding feature $\mathbf{X} \in \mathbb{R}^{|V| \times D}$ with $D$ the feature dimension, and $E = \{e_{ij}\}_{1 \leq i < j \leq |V|}$ is the set of edges. Let $\Gamma_v$ denote the set of neighborhood nodes around $v$, and $d_v$ the node degrees of $v$. We use $\mathbf{A}$ denote the adjacency matrix where $\mathbf{A}_{ij} = \mathbf{A}_{ji} = 1$ if $e_{ij} \in E$, and 0 elsewhere. $\mathbf{D} = \mathrm{diag}(d_v)$ denotes the degree matrix. When the context is clear, we simplify the notation $\Gamma_{v_i} \to \Gamma_i$, and the same manner for other symbols. For multi-layer GNNs, let $\mathbf{z}_v^{(k)}$ denote the hidden representation of node $v$ in the $k$-th layer, and with $\mathbf{z}_v^{(0)} = \mathbf{x}_v$ the initial node features. Simple Graph Convolution processes node representations as: $\mathbf{z}_v^{(k)} = \mathbf{W}^{(k)} \left( \sum_{u \in \Gamma_v \cup \{v\}} d_u^{-1/2} d_v^{-1/2} \mathbf{z}_u^{(k-1)} \right) + \mathbf{b}^{(k)}$, where $\mathbf{W}^{(k)}$ and $\mathbf{b}^{(k)}$ are trainable parameters in $k$-th layer. In transductive node classification, let $V_{\mathrm{train}} \subset V$ denote the set of $N$ training nodes associated with labels $y$. ERM of SGC in this task is $\hat{\theta} = \arg\min_{\theta \in \Theta} \frac{1}{N} \sum_{v \in V_{\mathrm{train}}} \ell(\mathbf{z}_v^{(k)}, y_v) + \frac{\lambda}{2} \|\theta\|_2^2$. Due to the linearity of SGC, parameters $\mathbf{W}^{(k)}$ and $\mathbf{b}^{(k)}$ in each layer can be unified, and predictions after $k$ layers can be simplified as $\mathbf{y} = \arg\max(\tilde{\mathbf{D}}^{-\frac{1}{2}} \tilde{\mathbf{A}} \tilde{\mathbf{D}}^{-\frac{1}{2}})^k \mathbf{X} \mathbf{W} + \mathbf{b}$ with $\tilde{\mathbf{A}} = \mathbf{A} + \mathbf{I}$ and $\tilde{\mathbf{D}}$ the degree matrix of $\tilde{\mathbf{A}}$. Therefore, for node representations $\mathbf{Z}^{(k)} = (\tilde{\mathbf{D}}^{-\frac{1}{2}} \tilde{\mathbf{A}} \tilde{\mathbf{D}}^{-\frac{1}{2}})^k \mathbf{X}$ with $\mathbf{y}$ and cross-entropy loss, $\ell(\cdot, \cdot)$ is convex. The parameters $\theta$ in $\ell$ consist of matrix $\mathbf{W} \in \mathbb{R}^{D \times |\mathrm{Class}|}$ and vector $\mathbf{b} \in \mathbb{R}^{|\mathrm{Class}|}$ with $|\mathrm{Class}|$ the number of class, and can be solved via logistic regression.

**Additional Notations**   In what follows, we shall build our influence analysis upon SGC. For notational simplification, we omit $(k)$ in $\mathbf{Z}^{(k)}$ and use $\mathbf{Z}$ to denote the last-layer node representations from SGC. We use $\mathcal{I}^*(-e_{ij}) = \hat{\theta}(-e_{ij}) - \hat{\theta}$ to denote the actual model parameters' change where $\hat{\theta}(-e_{ij})$ is obtained through ERM when $e_{ij}$ is removed from $E$. Likewise, $\mathcal{I}^*(-v_i)$ denotes the change from $v_i$'s removal from graph $G$. $\mathcal{I}(-e_{ij})$ and $\mathcal{I}(-v_i)$ are the corresponding estimated influence for $\mathcal{I}^*(-e_{ij})$ and $\mathcal{I}^*(-v_i)$ based on influence functions, respectively.

## 3   Modeling the Influence of Elements in Graphs

We mainly consider the use of influence functions of two fundamental operations over an attributed graph: removing an edge (in Section 3.1) and removing a complete node (in Section 3.2).

### 3.1   Influence of Edge Removal

With message passing through edges in graph convolution, removing an edge will incur representational changes in $\mathbf{Z}$. When $e_{ij}$ is removed, the changes come from two aspects: **(1)** The message passing for node features via the removed edge will be blocked, and all the representations of $k$-hop neighboring nodes of the removed edge will be affected. **(2)** Due to the normalization operation over $\mathbf{A}$, the degree of all adjacent edges $e_{jk}, \forall k \in \Gamma_i$ and $e_{ik}, \forall k \in \Gamma_j$ will be changed, and these edges will have a larger value in $\tilde{\mathbf{D}}^{-\frac{1}{2}} \tilde{\mathbf{A}} \tilde{\mathbf{D}}^{-\frac{1}{2}}$. We have the following expression to describe the representational changes $\Delta(-e_{ij})$ of node representations $\mathbf{Z}$ in SGC incurred by removing $e_{ij}$.

$$\Delta(-e_{ij}) = [(\tilde{\mathbf{D}}(-e_{ij})^{-\frac{1}{2}} \tilde{\mathbf{A}}(-e_{ij}) \tilde{\mathbf{D}}(-e_{ij})^{-\frac{1}{2}})^k - (\tilde{\mathbf{D}}^{-\frac{1}{2}} \tilde{\mathbf{A}} \tilde{\mathbf{D}}^{-\frac{1}{2}})^k] \mathbf{X}. \tag{2}$$

$\mathbf{A}(-e_{ij})$ is the modified adjacency matrix with $\mathbf{A}(-e_{ij})_{ij/ji} = 0$ and $\mathbf{A}(-e_{ij}) = \mathbf{A}$ elsewhere. $\tilde{\mathbf{A}}(-e_{ij}) = \mathbf{A}(-e_{ij}) + \mathbf{I}$ and $\tilde{\mathbf{D}}(-e_{ij})$ the degree matrix of $\tilde{\mathbf{A}}(-e_{ij})$. By having $\Delta(-e_{ij})$, we can access the change in every node. Let $\delta_k(-e_{ij})$ denotes the $k$-th row in $\Delta(-e_{ij})$. $\delta_k = 0$ implies no change in $k$-th node from removing $e_{ij}$, and $\delta_k \neq 0$ indicates a change in $\mathbf{z}_k$.

We proceed to use influence functions to characterize the counterfactual effect of removing $e_{ij}$. Our high-level idea is, from an influence functions perspective, representational changes in nodes $\mathbf{z} \to \mathbf{z} + \delta$ is equivalent to removing training instances with feature $\mathbf{z}$, and adding new training instances with feature $\mathbf{z} + \delta$ and with the same labels. The problem thus turns back to an instance reweighing problem developed by influence functions. In this case, we have the lemma below to prove the influence functions' linearity.

**Lemma 3.1.** *Consider empirical risk minimization* $\hat{\theta} = \arg\min_{\theta \in \Theta} \sum_i \ell(\mathbf{x}_i, y_i)$ *and* $\hat{\theta}(\mathbf{x}_j \to \mathbf{x}_j + \delta) = \arg\min_{\theta \in \Theta} \sum_{i \neq j} \ell(\mathbf{x}_i, y_i) + \ell(\mathbf{x}_j + \delta, y_j)$ *with some twice-differentiable and strictly convex*

$\ell$, let $\mathcal{I}^*(\mathbf{x}_j \to \mathbf{x}_j + \delta) = \hat{\theta}(\mathbf{x}_j \to \mathbf{x}_j + \delta) - \hat{\theta}$, the estimated influence satisfies linearity:

$$\mathcal{I}(\mathbf{x}_j \to \mathbf{x}_j + \delta) = \mathcal{I}(-\mathbf{x}_j) + \mathcal{I}(+(\mathbf{x}_j + \delta)). \tag{3}$$

By having Lemma 3.1, we are ready to derive a proposition from characterizing edge removal.

**Proposition 3.2.** *Let $\delta_k(-e_{ij})$ denote the $k$-th row of $\Delta(-e_{ij})$. The influence of removing an edge $e_{ij} \in E$ from graph $G$ can be estimated by:*

$$\mathcal{I}(-e_{ij}) = \mathcal{I}(\mathbf{z} \to \mathbf{z} + \delta(-e_{ij})) = \sum_k \mathcal{I}(+(\mathbf{z}_k + \delta_k(-e_{ij}))) + \mathcal{I}(-\mathbf{z}_k)$$

$$= -\mathbf{H}_{\hat{\theta}}^{-1} \sum_{v_k \in V_{train}} (\nabla_{\hat{\theta}} \ell(\mathbf{z}_k + \delta_k(-e_{ij}), y_k) - \nabla_{\hat{\theta}} \ell(\mathbf{z}_k, y_k)). \tag{4}$$

*Proof.* The second equality comes from Lemma 3.1, and the third equality comes from Equation (1). Realize that removing two representations $\mathcal{I}(-z_i, -z_j) = \mathcal{I}(-z_i) + \mathcal{I}(-z_j)$ completing the proof. $\square$

Proposition 3.2 offers an approach to calculate the estimated influence of removing $e_{ij}$. In practice, having the inverse hessian matrix, a removal only requires users to compute the updated gradients $\nabla_{\hat{\theta}} \ell(\mathbf{z}_k + \delta_k(-e_{ij}), y_k)$ and its original gradients for all affected nodes in $(k+1)$-hop neighbors.

### 3.2 INFLUENCE OF NODE REMOVAL

We address the case of node removal. The impact from removing a node $v_i$ from graph $G$ to parameters' change are two-folds: **(1)** The loss term $\ell(\mathbf{x}_i, y_i)$ will no longer involved in ERM if $v_i \in V_{\text{train}}$. **(2)** All edges link to this node $\{e_{ij}\}, \forall j \in \Gamma_i$ will be removed either. The first aspect can be deemed as a regular training instance removal similar to an i.i.d. case, and the second aspect be can an incremental extension from edge removal in Proposition 3.2.

The representational changes from removing node $v_i$ can be expressed as:

$$\Delta(-v_i) = [(\tilde{\mathbf{D}}(-v_i)^{-\frac{1}{2}} \tilde{\mathbf{A}}(-v_i) \tilde{\mathbf{D}}(-v_i)^{-\frac{1}{2}})^k - (\tilde{\mathbf{D}}^{-\frac{1}{2}} \tilde{\mathbf{A}} \tilde{\mathbf{D}}^{-\frac{1}{2}})^k] \mathbf{X}, \tag{5}$$

with $\mathbf{A}(-v_i)_{jk/kj} = \mathbf{A}_{jk/kj}, \forall j, k : j \neq i \wedge k \notin \Gamma_i$, and $\mathbf{A}(-v_i) = 0$ elsewhere. Similarly, $\tilde{\mathbf{A}}(-v_i) = \mathbf{A}(-v_i) + \mathbf{I}$ and $\tilde{\mathbf{D}}(-v_i)$ is the corresponding degree matrix of $\tilde{\mathbf{A}}(-v_i)$. Having $\Delta(-v_i)$, Lemma 3.1 and Proposition 3.2, we state the estimated influence of removing $v_i$.

**Proposition 3.3.** *Let $\delta_j(-v_i)$ denote the $j$-th row of $\Delta(-v_i)$. The influence of removing node $v_i$ from graph $G$ can be estimated by:*

$$\mathcal{I}(-v_i) = \mathcal{I}(-\mathbf{z}_i) + \mathcal{I}(\mathbf{z} \to \mathbf{z} + \delta(-v_i)) = \mathcal{I}(-\mathbf{z}_i) + \sum_j \mathcal{I}(+(\mathbf{z}_j + \delta_j(-v_i))) + \mathcal{I}(-\mathbf{z}_j)$$

$$= -\mathbb{1}_{v_i \in V_{train}} \cdot \mathbf{H}_{\hat{\theta}}^{-1} \nabla_{\hat{\theta}} \ell(\mathbf{z}_i, y_i) - \mathbf{H}_{\hat{\theta}}^{-1} \sum_{v_j \in V_{train}} (\nabla_{\hat{\theta}} \ell(\mathbf{z}_j + \delta_j(-v_i), y_j) - \nabla_{\hat{\theta}} \ell(\mathbf{z}_j, y_j)), \tag{6}$$

*where $\mathbb{1}$ is an indicator function.*

*Proof.* Combining Lemma 3.1 and Equation (1) completes the proof. $\square$

## 4 THEORETICAL ERROR BOUNDS

In the above section, we show how to estimate the changes of model parameters due to edge removal: $\hat{\theta} \to \hat{\theta}(-e_{ij})$ and node removals: $\hat{\theta} \to \hat{\theta}(-v_i)$. In this section, we study the error between the estimated influence given by influence functions $\mathcal{I}$ and the actual influence $\mathcal{I}^*$ obtained by model retraining. We give upper error bounds on edge removal $\|\mathcal{I}^*(-e_{ij}) - \mathcal{I}(-e_{ij})\|_2$ (see Theorem 4.1) and node removal $\|\mathcal{I}^*(-v_i) - \mathcal{I}(-v_i)\|_2$ (see Corollary A.1).

In what follows, we shall assume the second derivative of $\ell(\cdot, \cdot)$ is Lipschitz continuous at $\theta$ with constant $C$ based on the convergence theory of Newton's method. To simplify the notations, we use $\mathbf{z}_i' = \mathbf{z}_i + \delta_i$ to denote the new representation of $v_i$ obtained after removing an edge or a node, where $\delta_i$ is the row vector of $\Delta(-e_{ij})$ or $\Delta(-v_i)$ depending on the context.

**Theorem 4.1.** *Let $\sigma_{min} \geq 0$ denote the smallest eigenvalue of all eigenvalues of Hessian matrices $\nabla^2_{\hat{\theta}}\ell(\mathbf{z}_i, y_i), \forall v_i \in V_{train}$ of the original model $\hat{\theta}$. Let $\sigma'_{min} \geq 0$ denote the smallest eigenvalue of all eigenvalues of Hessian matrices $\nabla^2_{\hat{\theta}(-e_{ij})}\ell(\mathbf{z}_i, y_i), \forall v_i \in V_{train}$ of the retrained model $\hat{\theta}(-e_{ij})$ with $e_{ij}$ removed from graph G. Use L denote the set $\{v : \mathbf{z}' \neq \mathbf{z}\}$ containing affected nodes from the edge removal, and $Err(-e_{ij}) = \|\mathcal{I}^*(-e_{ij}) - \mathcal{I}(-e_{ij})\|_2$. Recall $\lambda$ is the $\ell_2$ regularization strength, we have an upper bound on the estimated error of model parameters' change:*

$$Err(-e_{ij}) \leq \frac{N^3 C}{(N\lambda + (N - |L|)\sigma_{min} + \sigma'_{min}|L|)^3} \cdot \|\sum_{v_l \in L}(\nabla_{\hat{\theta}}\ell(\mathbf{z}'_l, y_l) - \nabla_{\hat{\theta}}\ell(\mathbf{z}_l, y_l))\|^2_2$$
$$+ \frac{N}{N\lambda + (N - |L|)\sigma_{min} + \min(\sigma_{min}, \sigma'_{min})|L|} \cdot \|\sum_{v_l \in L}(\nabla_{\hat{\theta}}\ell(\mathbf{z}'_l, y_l) - \nabla_{\hat{\theta}}\ell(\mathbf{z}_l, y_l))\|_2. \tag{7}$$

*Proof sketch.* We use the one-step Newton approximation (Pregibon, 1981) as an intermediate step to derive the bound. The first term is the difference between the actual change $\mathcal{I}^*(-e_{ij})$ and its Newton approximation, and the second term is the difference between the Newton approximation and the estimated influence $\mathcal{I}(-e_{ij})$. Combining these two parts result the bound. □

**Remark 4.2.** We have the following main observations from Theorem 4.1. **(1)** The estimation error of influence function is controlled by the $\ell_2$ regularization strength within a factor of $\mathcal{O}(1/\lambda)$. A stronger regularization will likely produce a better approximation. **(2)** The error is controlled by the inherent property of a model. A smoother model in terms of its hessian matrix will help lower the upper bound. **(3)** The upper bound is controlled by the norm of the changed gradient from $\mathbf{z} \rightarrow \mathbf{z}'$. Intuitively, if removing $e_{ij}$ incurs smaller changes in node representations, the approximation of the actual influence would be more accurate. Also, a smaller $Err(-v_i)$ is expected if the model is less prone to changes in training samples. **(4)** There are no significant correlations between the bound and the number of training nodes $N$. As a special case, if $\sigma_{min} = \sigma'_{min} = 0$, the bound is irrelevant to $N$. We attach empirical verification for our bound in Appendix D.

Similar to Theorem 4.1, we have Corollary A.1 to derive an upper bound on $\|\mathcal{I}^*(-v_i) - \mathcal{I}(-v_i)\|_2$ for removing a node $v_i$ from graph presented in Appendix A.

## 5 EXPERIMENTS

We conducted three major experiments: **(1)** Validate the estimation accuracy of our influence functions on graph in Section 5.2; **(2)** Utilize the estimated edge influence to carry out adversarial attacks and graph rectification for increasing model performance in Section 5.3; and **(3)** Utilize the estimated node influence to carry out adversarial attacks on GCN (Kipf & Welling, 2017) in Section 5.4.

### 5.1 SETUP

We choose six real-world graph datasets:*Cora*, *PubMed*, *CiteSeer* (Sen et al., 2008), *WiKi-CS* (Mernyei & Cangea, 2020), *Amazon Computers*, and *Amazon Photos* (Shchur et al., 2018) in our experiments. Statistics of these datasets are outlined in Appendix B Table 4. For the *Cora*, *PubMed*, and *CiteSeer* datasets, we used their public train/val/test splits. For the Wiki-CS datasets, we took a random single train/val/test split provided by Mernyei & Cangea (2020). For the Amazon datasets, we randomly selected 20 nodes from each class for training, 30 nodes from each class for validation and used the rest nodes in the test set. All the experiments are conducted under the transductive node classification settings. We only use the last three datasets for influence validation.

### 5.2 VALIDATING INFLUENCE FUNCTIONS ON GRAPHS

**Validating Estimated Influence**   We compared the estimated influence of removing a node/edge with its corresponding ground truth effect. The actual influence is obtained by re-training the model after removing a node/edge and calculating the change in the total cross-entropy loss. We also validated the estimated influence of removing node embeddings, for example, removing $\ell(\mathbf{z}_i, y_i)$ of node $v_i$ from the ERM objective while keeping the embeddings of other nodes intact. Figure 2

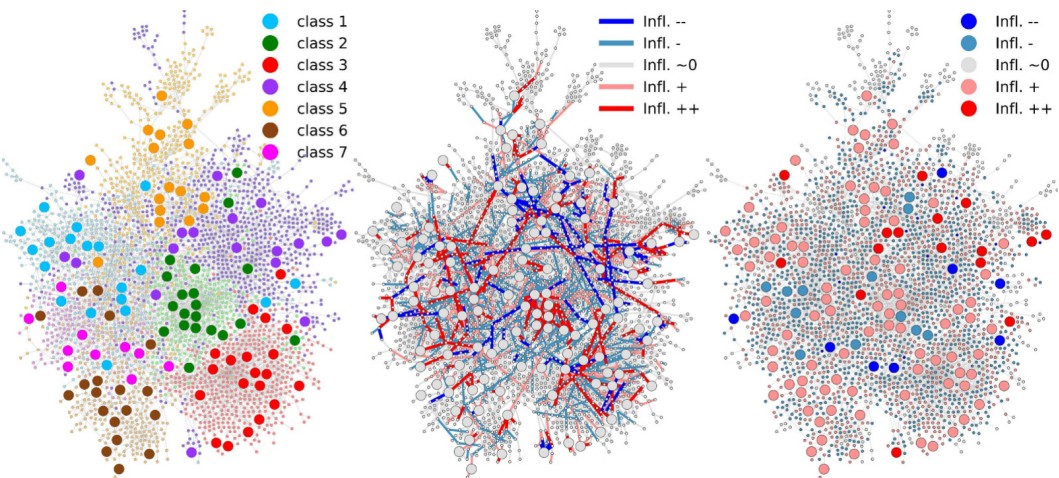

Figure 1: The *Cora* experiment – the estimated influences of individual training nodes/edges on the validation loss. The largest connected component of the *Cora* dataset is visualized here. **Left**: The dataset. The node size indicates if a node is in the training subset (large) or not (small). **Middle**: Influence of the training edges. Each edge is colored accordingly to its estimated influence value (*blue* - negative influence, removing it is expected to decrease the loss on the validation set; *red* – positive influence, removing it is expected to increase the loss on the validation set; and grey – little influence. The deeper color indicates higher influence.). **Right**: Influence of the training nodes. The same color scheme in the middle plot is used here.

shows that the estimated influence correlates highly with the actual influence (Spearman correlation coefficients range from 0.847 to 0.981). More results are included in Figure 4 in the appendix.

**Visualization** Figure 1 visualizes the estimated influence of edge and node removals on the validation loss for the *Cora* dataset. This visualization hints at opportunities for improving the test performance of a model or attacking a model by removing nodes/edges with noticeable influences (see experiments in Sections 5.3 and 5.4).

## 5.3 APPLICATIONS OF THE ESTIMATED EDGE INFLUENCE

The estimated influence of edge removals on the validation set can be utilized to improve the test performance of SGC or carry out adversarial attacks on SGC/GCN.

**Graph Rectification via Edge Removals** We begin by investigating the impact of edges with negative influences. Based on our influence analysis, removing negative influence edges from the original will decrease validation loss. Thus the classification accuracy on the test set is expected to increase correspondingly. We sort the edges by their estimated influences in descending order, then cumulatively remove edges starting from the one with the lowest negative influence. We train the SGC model, fine-tune it on the public split validation set and select the number of negative influence edges to be

Table 1: Our performance via eliminating edges with negative influence values.

| Methods | *Cora* | *Pubmed* | *Citeseer* |
|---|---|---|---|
| GCN | $81.4 \pm 0.4$ | $79.0 \pm 0.4$ | $70.1 \pm 0.5$ |
| GAT | $\mathbf{83.3} \pm 0.7$ | $78.5 \pm 0.3$ | $72.6 \pm 0.6$ |
| FGCN | $79.8 \pm 0.3$ | $77.4 \pm 0.3$ | $68.8 \pm 0.6$ |
| GIN | $77.6 \pm 1.1$ | $77.0 \pm 1.2$ | $66.1 \pm 0.9$ |
| DGI | $82.5 \pm 0.7$ | $78.4 \pm 0.7$ | $71.6 \pm 0.7$ |
| SGC | $81.0 \pm 0.0$ | $78.9 \pm 0.0$ | $71.9 \pm 0.1$ |
| Ours | $81.8 \pm 0.0$ | $\mathbf{79.7} \pm 0.0$ | $\mathbf{73.7} \pm 0.0$ |

removed by validation accuracy. For a fair comparison, we fix the test set remaining unchanged regarding the removal of the edges. The results are derived based on Figure 8 and displayed in Table 1, where we also report the performance of several classical and state-of-the-art GNN models on the original whole set as references, including GCN (Kipf & Welling, 2017), GAT (Veličković et al., 2018), FGCN (Chen et al., 2018), GIN (Xu et al., 2019b), DGI (Velickovic et al., 2019) with a nonlinear activation function and SGC (Wu et al., 2019).

We demonstrate that our proposed method can marginally improve the accuracy of SGC from the data perspective and without any change to the original model structure of SGC, which validates

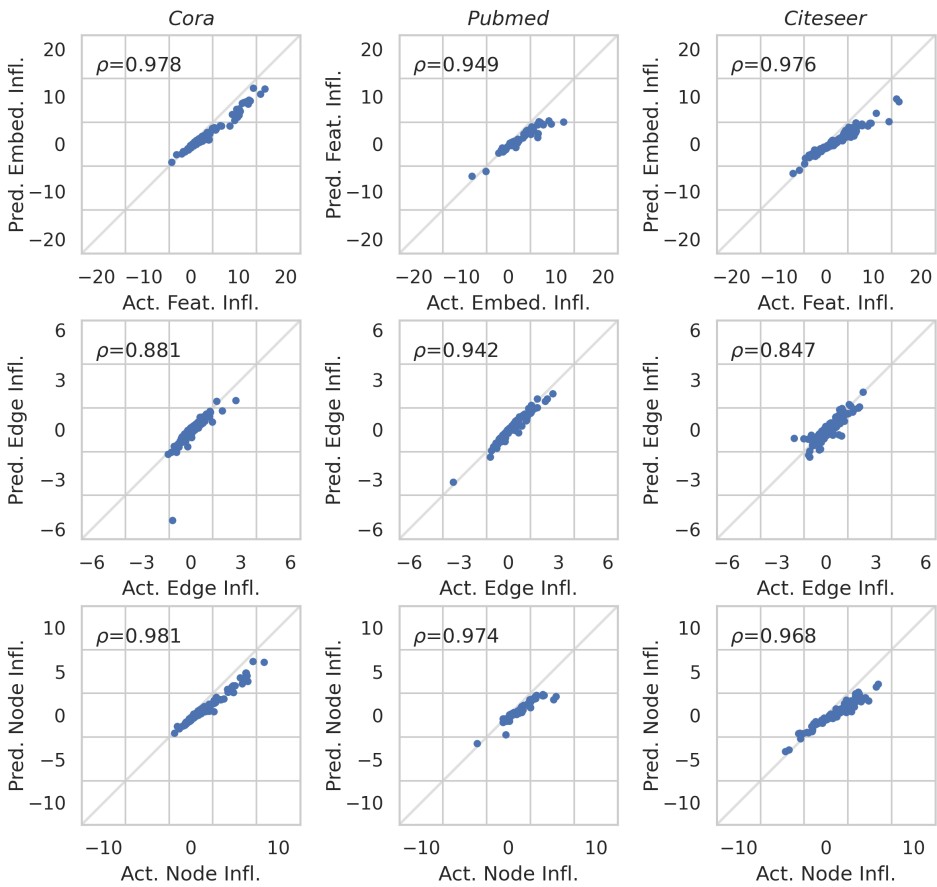

Figure 2: Estimated influence vs. actual influence. Three datasets are used in this illustration *Cora* (left column), *Pubmed* (middle column) and *Citeseer* (right column). In all plots, the horizontal axes indicate the actual influence on the validation set, the vertical axes indicate the predicted influence, and $\rho$ indicates Spearman's correlation coefficient between our predictions and the actual influences. **Top row**: Influence of node embedding removal. Each point represents a training node embedding **Middle row**: Influence of edge removals. Each point corresponds to a removed edge. **Bottom row**: Influence of node removal. Each point represents a removed training node.

Table 2: Grey-box attacks to GCN via edge removals. A lower performance indicates a more successful attack. The best attacks are in **bold** font. The number following the dataset name is the pre-attack performance. '-' denotes an out-of-memory issue encountered on GPU with 24GB VRAM.

| Dataset | *Cora - 81.10%* | | | *Citeseer - 70.07%* | | | *Pubmed - 79.80%* | | |
|---|---|---|---|---|---|---|---|---|---|
| Elimination Rate | 1% | 3% | 5% | 1% | 3% | 5% | 1% | 3% | 5% |
| DICE | 79.9% | 80.1% | 80.0% | 71.1% | 70.3% | 69.8% | 79.4% | 79.7% | 79.1% |
| GraphPoison | 80.0% | 80.1% | 79.6% | 70.2% | 70.1% | 70.0% | 79.4% | 79.7% | 79.1% |
| MetaAttack | 79.6% | 77.1% | 73.3% | 70.4% | 69.3% | 65.4% | - | - | - |
| Ours | **77.3%** | **74.2%** | **72.8%** | **69.3%** | **67.4%** | **64.7%** | **69.3%** | **65.2%** | **64.1%** |

the impacts of edges with negative influences. In addition, the performance of the SGC model with eliminating the negative influence edges can outperform other GNN-based methods in most cases.

**Attacking SGC via Edge Removals**  We investigated how to use edge removals to deteriorate SGC performance. Based on the influence analysis, removing an edge with a positive estimated influence can increase the model loss and decrease the model performance on the validation set. Thus, our attack is carried out in the following way. We first calculated the estimated influence of all edges and cumulatively removed edges with the highest positive influence one at a time. Every time we remove a new edge, we retrain the model to obtain the current model performance. We remove 100 edges in total for each experiment.

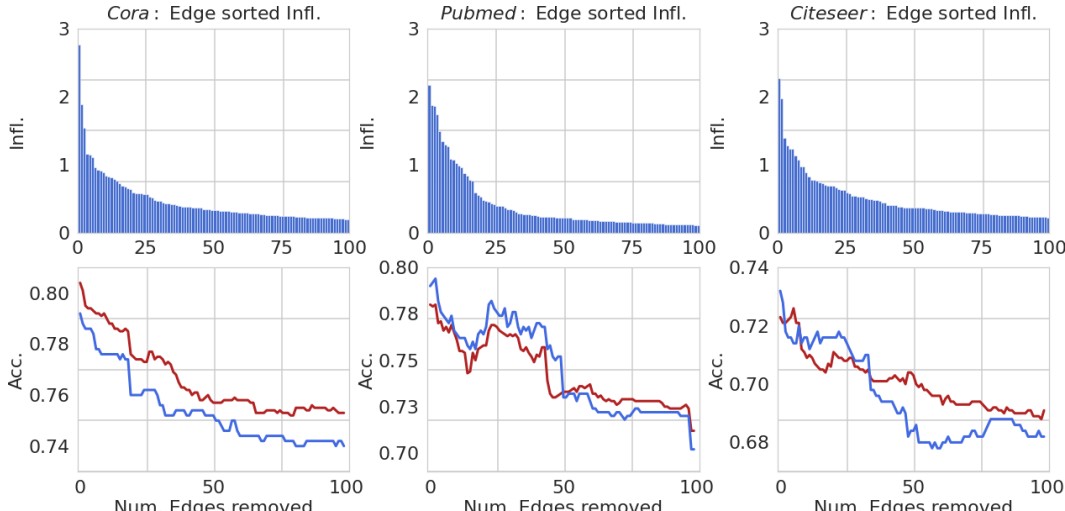

Figure 3: Study of edges with positive influence on both validation (blue) and test (red) set. Columns correspond to *Cora*, *Pubmed* and *Citeseer* datasets. **Top**: scale of values of the edges with negative influence. **Bottom**: accuracy drop by cumulatively removing edges with positive influence.

Table 3: Performance of node removing attack. Lower performance means better attacks. The number after the dataset name means the performance of GCN model without an attack. Victim model's test accuracy averaged over 25 runs on the citation network.

| Dataset | Cora - 81.10% | | | Citeseer - 70.07% | | | Pubmed - 79.80% | | |
|---|---|---|---|---|---|---|---|---|---|
| Removing Rate | 5% | 10% | 15% | 5% | 10% | 15% | 5% | 10% | 15% |
| Random | 80.4% | 80.3% | 80.2% | 70.6% | 69.0% | 69.2% | 78.9% | 79.6% | 77.3% |
| Degree | 80.3% | 78.7% | 79.0% | 69.4% | 68.3% | 68.4% | 79.1% | 79.6% | 77.4% |
| Ours | **74.7%** | **59.8%** | **57.9%** | **69.5%** | **65.5%** | **56.1%** | **79.0%** | **77.2%** | **75.2%** |

We present our results in Figure 3. Apparently, in general, the accuracy of SGC on node classification drops significantly. We notice the influence of edges is approximately power-law distributed, where only a small proportion of edges has a relatively significant influence. The performance worsens with increasingly cumulative edge removals on both validation and test sets. The empirical results verify our expectations of edges with a positive estimated influence.

**Attacking GCN via Surrogate SGC**   We further explored the impact of removing positive influences edges under adversarial grey-box attack settings. Here, we followed Zügner & Günnemann (2019) to interpret SGC as a surrogate model for attacking the GCN (Kipf & Welling, 2017) as a victim model, where the assumption lays under that the increase of loss on SGC can implicitly drop the performance of GCN. We eliminated positive influence edges at different rates $1\%, 3\%, 5\%$ among all edges. The drop in accuracy was compared against DICE (Zügner et al., 2018), Graph-Poison (Bojchevski & Günnemann, 2019), MetaAttack (Zügner et al., 2018). For a fair comparison, we restrict the compared method can only perturb graph structures via edge removals.

Our results are presented in Table 2. Our attack strategy achieves the best performance in all the scenarios of edge eliminations, especially on *Pubmed* with 1% edge elimination rate. Our attack model outperforms others by over 10% in accuracy drop. Since we directly estimate the impact of edges on the model parameter change, our attack strategy is more effective in seeking the most vulnerable edges to the victim model. These indicate that our proposed influence on edges can guide the construction of grey-box adversarial attacks on graph structures.

## 5.4   INFLUENCE OF NODE REMOVAL

**Attacking GCN via Node Removals**   In this section, we study the impact of training nodes with a positive influence on transductive node classification tasks. Again, we assume that eliminating the positive influence nodes derived from SGC may implicitly harm the GCN model. We sort the nodes

by our estimated influence in descending order and cumulatively remove the nodes from the training set. We built two baseline methods, Random and Degree, to compare the accuracy drop in different node removal ratios: $5\%, 10\%, 15\%$. For the Random baseline, we randomly remove the nodes from the training sets. For Degree baseline, we remove nodes by their degree in descending order.

According to Table Table 3, the model performance on GCN drops by a large margin in all three citation network datasets as the selected positive influence node is removed, especially on the Cora dataset. The model outperforms the baseline over $20\%$ on $15\%$ removing ratio. These results indicate that our estimation of node influence can be used to guide the adversarial attack on GCN in the settings of node removal.

## 6 RELATED WORKS

**Influence Functions**  Recently, more efforts have been dedicated to investigating influence functions (Koh et al., 2019; Giordano et al., 2019; Ting & Brochu, 2018) in various applications, such as,computer vision (Koh & Liang, 2017), natural language processing (Han et al., 2020), tabular data (Wang et al., 2020b), causal inference (Alaa & Van Der Schaar, 2019), data poisoning attack (Fang et al., 2020; Wang et al., 2020a), and algorithmic fairness (Li & Liu, 2022). In this work, we propose a major extension of influence functions to graph-structured data and systemically study how we can estimate the influence of nodes and edges in terms of different editing operations on graphs. We believe our work complements the big picture of influence functions in machine learning applications.

**Understanding Graph Data**  Besides influence functions, there are many other approaches to exploring the underlying patterns in graph data and its elements. Explanation models for graphs (Ying et al., 2019; Huang et al., 2022; Yuan et al., 2021; Bajaj et al., 2021; Abrate & Bonchi, 2021) provide an accessible relationship between the model's predictions and corresponding elements in graphs or subgraphs. They show how the graph's local structure or node features impact the decisions from GNNs. As a major difference, these approaches tackle model inference with fixed parameters, while we focus on a counterfactual effect and investigate the contributions from the presence of nodes and edges in training data to decisions of GNN models in the inference stage.

**Adversarial Attacks on Graph**  The adversarial attack on an attributed graph is usually conducted by adding perturbations on the graphic structure or node features (Zügner & Günnemann, 2019; Zheng et al., 2021). In addition, Zhang et al. (2020) introduces an adversarial attack setting by flipping a small fraction of node labels in the training set that causes a significant drop in model performance. A majority of the attacker models (Zügner et al., 2018; Xu et al., 2019a) on graph structure are constructed based on the gradient information on both edges and node features and achieved costly but effective attacking results. These attacker models rely mainly on greedy-based methods to find the graph structure's optimal perturbations. We only focus on the perturbations resulting from eliminating edges and directly estimate the change of loss in response to the removal effect guided by the proposed influence-based approach.

## 7 CONCLUSIONS

We have developed a novel influence analysis to understand the effects of graph elements on the parameter changes of GCNs without needing to retrain the GCNs. We chose Simple Graph Convolution due to its convexity and its competitive performance to non-linear GNNs on a variety of tasks. Our influence functions can be used to approximate the changes in model parameters caused by edge or node removals from an attributed graph. Moreover, we provided theoretical bounds on the estimation error of the edge and node influence on model parameters. We experimentally validated the accuracy and effectiveness of our influence functions by comparing its estimation with the actual influence obtained by model retraining. We showed in our experiments that our influence functions could be used to reliably identify edge and node with negative and positive influences on model performance. Finally, we demonstrated that our influence function could be applied to graph rectification and model attacks.

ACKNOWLEDGEMENT

We would like to thank the three anonymous reviewers for their constructive questions and invaluable suggestions. This work is partially supported by NSF DMR 1933525 and NSF OAC 1920147. Any opinions or conclusions in this paper are those of the authors and do not reflect the views of the funding agencies.

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

# A    PROOFS

**Lemma 3.1.** *Consider empirical risk minimization $\hat{\theta} = \arg\min_{\theta \in \Theta} \sum_i \ell(\mathbf{x}_i, y_i)$ and $\hat{\theta}(\mathbf{x}_j \to \mathbf{x}_j + \delta) = \arg\min_{\theta \in \Theta} \sum_{i \neq j} \ell(\mathbf{x}_i, y_i) + \ell(\mathbf{x}_j + \delta, y_j)$ with some twice-differentiable and strictly convex $\ell$, let $\mathcal{I}^*(\mathbf{x}_j \to \mathbf{x}_j + \delta) = \hat{\theta}(\mathbf{x}_j \to \mathbf{x}_j + \delta) - \hat{\theta}$, the estimated influence satisfies linearity:*

$$\mathcal{I}(\mathbf{x}_j \to \mathbf{x}_j + \delta) = \mathcal{I}(-\mathbf{x}_j) + \mathcal{I}(+(\mathbf{x}_j + \delta)). \tag{3}$$

*Proof.* Notice the actual model parameters in response of the perturbations $\Delta$ can be denoted as:

$$\hat{\theta}(\mathbf{x}_j \to \mathbf{x}_j + \delta) \overset{\text{def}}{=} \arg\min_{\theta \in \Theta} \frac{1}{N} \sum_{k=1}^{N} \ell(\mathbf{x}_k, y_k) - \frac{1}{N}\ell(\mathbf{x}_j, y_j) + \frac{1}{N}\ell(\mathbf{x}_j + \delta, y_j)$$

In this case, the actual change in model parameters in response of the perturbations can be represented as: $\mathcal{I}(\mathbf{x}_j \to \mathbf{x}_j + \delta) = \hat{\theta}(\mathbf{x}_j \to \mathbf{x}_j + \delta) - \hat{\theta}$. For estimating $\Delta\theta$, we start by considering the parameter change from up weighting infinite small $\varepsilon$ on $\{\mathbf{x}'_l\}$ and down weight infinite small $\varepsilon$ on $\{\mathbf{x}_l\}$ where $\forall l \in L$. By definition, the model parameter in response of perturbation $\hat{\theta}_\varepsilon$ can be represented as:

$$\hat{\theta}_\varepsilon \overset{\text{def}}{=} \arg\min_{\theta \in \Theta} \frac{1}{N} \sum_{k=1}^{N} \ell(\mathbf{x}_k, y_k) - \varepsilon\ell(\mathbf{x}_j, y_j) + \varepsilon\ell(\mathbf{x}_j + \delta, y_j) \tag{8}$$

The change of model parameter due to the modification on group of data's weight on loss be:

$$\Delta\theta_\varepsilon = \hat{\theta}_\varepsilon - \hat{\theta} \tag{9}$$

Since $\hat{\theta}_\varepsilon$ minimize the changed loss function under perturbation, take the derivative:

$$\begin{aligned} 0 = &\frac{1}{N} \sum_{k=1}^{N} \nabla_{\hat{\theta}_\varepsilon} \ell(\mathbf{x}_k, y_k) - \varepsilon\nabla_{\hat{\theta}_\varepsilon} \ell(\mathbf{x}_j, y_j) \\ &+ \varepsilon\nabla_{\hat{\theta}_\varepsilon} \ell(\mathbf{x}_j + \delta, y_j) \end{aligned} \tag{10}$$

Apply the first order Taylor expansion of $\hat{\theta}_\varepsilon$ on $\hat{\theta}$ on the right side of the equation, we have:

$$\begin{aligned} 0 = &\left[ \frac{1}{N} \sum_{k=1}^{N} \nabla_\theta \ell(\mathbf{x}_k, y_k) + \varepsilon\nabla_\theta \ell(\mathbf{x}_j + \delta, y_j) - \varepsilon\nabla_\theta \ell(\mathbf{x}_j, y_j) \right] + \\ &\left[ \frac{1}{N} \sum_{k=1}^{N} \nabla_\theta^2 \ell(\mathbf{x}_k, y_k) + \varepsilon\nabla_\theta^2 \ell(\mathbf{x}_j + \delta, y_j) - \varepsilon\nabla_\theta^2 \ell(\mathbf{x}_j, y_j) \right] \cdot \Delta\theta_\varepsilon + o(\Delta\theta_\varepsilon^2) \end{aligned} \tag{11}$$

Since $\hat{\theta}$ minimize the loss function without perturbation, $\frac{1}{N} \sum_{k=1}^{N} \nabla_{\hat{\theta}_\varepsilon} \ell(\mathbf{x}_k, y_k) = 0$. Dropping $o(\Delta\theta_\varepsilon^2)$ term, We have:

$$\begin{aligned} \Delta\theta_\varepsilon \approx -&\left[ \frac{1}{N} \sum_{k=1}^{N} \nabla_\theta^2 \ell(\mathbf{x}_k, y_k) + \varepsilon\nabla_\theta^2 \ell(\mathbf{x}_j + \delta, y_j) - \varepsilon\nabla_\theta^2 \ell(\mathbf{x}_j, y_j) \right]^{-1} \cdot \\ &[\varepsilon\nabla_\theta \ell(\mathbf{x}_j + \delta, y_j) - \varepsilon\nabla_\theta \ell(\mathbf{x}_j, y_j)] \end{aligned} \tag{12}$$

Take the derivative of $\Delta\theta_\varepsilon$ over $\varepsilon$, by dropping $O(\varepsilon)$ terms we have:

$$\begin{aligned} \frac{\partial \Delta\theta_\varepsilon}{\partial \varepsilon} &= -\frac{1}{N} \sum_{k=1}^{N} \nabla_{\hat{\theta}}^2 \ell(\mathbf{x}_k, y_k)^{-1} \left[ \nabla_{\hat{\theta}} \ell(\mathbf{x}_j + \delta, y_j) - \nabla_{\hat{\theta}} \ell(\mathbf{x}_j, y_j) \right] \\ &= -H_{\hat{\theta}}^{-1} \sum_{l \in L} (\nabla\ell(\mathbf{x}_j + \delta, y_l) - \nabla\ell(\mathbf{x}_j, y_j)) \end{aligned} \tag{13}$$

For sufficient large $N$, by setting $\varepsilon$ to $\frac{1}{N}$, the changed we can approximate the actual change in model parameters using: $\mathcal{I}(\mathbf{x}_j \to \mathbf{x}_j + \delta) = \hat{\theta}(\mathbf{x}_j \to \mathbf{x}_j + \delta) - \hat{\theta} \approx \hat{\theta}_\varepsilon - \hat{\theta}$. Plugging in to Eq. (13) we finish the proof:

$$
\begin{aligned}
\mathcal{I}(\mathbf{x}_j \to \mathbf{x}_j + \delta) &\approx -H_{\hat{\theta}}^{-1}(\nabla \ell\,(\mathbf{x}_j + \delta, y_l) - \nabla \ell\,(\mathbf{x}_j, y_j)) \\
&= -H_{\hat{\theta}}^{-1} \nabla \ell\,(\mathbf{x}_j + \delta, y_l) + H_{\hat{\theta}}^{-1} \nabla \ell\,(\mathbf{x}_j, y_l) \\
&= \mathcal{I}(+(\mathbf{x}_j + \delta)) + \mathcal{I}(-\mathbf{x}_j).
\end{aligned}
\tag{14}
$$

$\square$

**Proposition 3.3.** *Let $\delta_j(-v_i)$ denote the $j$-th row of $\Delta(-v_i)$. The influence of removing node $v_i$ from graph $G$ can be estimated by:*

$$
\mathcal{I}(-v_i) = \mathcal{I}(-\mathbf{z}_i) + \mathcal{I}(\mathbf{z} \to \mathbf{z} + \delta(-v_i)) = \mathcal{I}(-\mathbf{z}_i) + \sum_j \mathcal{I}(+(\mathbf{z}_j + \delta_j(-v_i))) + \mathcal{I}(-\mathbf{z}_j)
$$

$$
= -\mathbb{1}_{v_i \in V_{train}} \cdot \mathbf{H}_{\hat{\theta}}^{-1} \nabla_{\hat{\theta}} \ell\,(\mathbf{z}_i, y_i) - \mathbf{H}_{\hat{\theta}}^{-1} \sum_{v_j \in V_{train}} (\nabla_{\hat{\theta}} \ell(\mathbf{z}_j + \delta_j(-v_i), y_j) - \nabla_{\hat{\theta}} \ell(\mathbf{z}_j, y_j)),
\tag{6}
$$

*where $\mathbb{1}$ is an indicator function.*

*Proof.* Similarly to the edge removal, we first calculate the node representation change incurred from the removal of the node $v_i$ of a 2-layer SGC as follow:

$$
\Delta(-v_i) = \left[ (\mathbf{D}_{-v_i}^{-\frac{1}{2}} \mathbf{A}_{-v_i} \mathbf{D}_{-v_i}^{-\frac{1}{2}})^2 - (\mathbf{D}^{-\frac{1}{2}} \mathbf{A} \mathbf{D}^{-\frac{1}{2}})^2 \right] \mathbf{X}.
\tag{15}
$$

The above change will affect a set of nodes, including the node $v_i$ itself and the 2-hop neighbors of the node $v_i$ connected neighbors. A set of nodes $S = \{s | s \in \mathcal{N}_i \cup_{j \in \mathcal{N}_i} \mathcal{N}_j\}$ capture the changed node embeddings in the training set, *i.e.*, $\delta_s \neq 0$, where $\Delta_{-v_i} = \{\delta_i\}_{i=1}^N$ in Eq. (15). The model parameter change of the removal of the node $v_i$ can be characterized by removing the representation of the node $v_i$ if the node $v_i$ is a training sample, and the node representation change from the set $S$. Thus, we have

$$
\begin{aligned}
\mathcal{I}(-v_i) &= -\mathbb{1}_{v_i \in V_{train}} \cdot \mathcal{I}\,(\mathbf{z}_i, y_i) + \sum_{s \in \{S \setminus v_i\}} (\mathcal{I}\,(\mathbf{z}_s', y_s) - \mathcal{I}\,(\mathbf{z}_s, y_s)) \\
&= -\mathbb{1}_{v_i \in V_{train}} \cdot \mathbf{H}_{\hat{\theta}}^{-1} \nabla \mathcal{L}_{CE}\,(\mathbf{z}_i, y_i) - \mathbf{H}_{\hat{\theta}}^{-1} \sum_{s \in S \setminus v_i} (\nabla \mathcal{L}_{CE}\,(\mathbf{z}_s', y_s) - \nabla \mathcal{L}_{CE}\,(\mathbf{z}_s, y_s)).
\end{aligned}
\tag{16}
$$

We finish the proof. $\square$

**Theorem 4.1.** *Let $\sigma_{min} \geq 0$ denote the smallest eigenvalue of all eigenvalues of Hessian matrices $\nabla_{\hat{\theta}}^2 \ell(\mathbf{z}_i, y_i), \forall v_i \in V_{train}$ of the original model $\hat{\theta}$. Let $\sigma_{min}' \geq 0$ denote the smallest eigenvalue of all eigenvalues of Hessian matrices $\nabla_{\hat{\theta}(-e_{ij})}^2 \ell(\mathbf{z}_i, y_i), \forall v_i \in V_{train}$ of the retrained model $\hat{\theta}(-e_{ij})$ with $e_{ij}$ removed from graph $G$. Use $L$ denote the set $\{v : \mathbf{z}' \neq \mathbf{z}\}$ containing affected nodes from the edge removal, and $Err(-e_{ij}) = \|\mathcal{I}^*(-e_{ij}) - \mathcal{I}(-e_{ij})\|_2$. Recall $\lambda$ is the $\ell_2$ regularization strength, we have an upper bound on the estimated error of model parameters' change:*

$$
\begin{aligned}
Err(-e_{ij}) &\leq \frac{N^3 C}{(N\lambda + (N - |L|)\sigma_{min} + \sigma_{min}'|L|)^3} \cdot \| \sum_{v_l \in L} (\nabla_{\hat{\theta}} \ell(\mathbf{z}_l', y_l) - \nabla_{\hat{\theta}} \ell(\mathbf{z}_l, y_l)) \|_2^2 \\
&+ \frac{N}{N\lambda + (N - |L|)\sigma_{min} + \min(\sigma_{min}, \sigma_{min}')|L|} \cdot \| \sum_{v_l \in L} (\nabla_{\hat{\theta}} \ell(\mathbf{z}_l', y_l) - \nabla_{\hat{\theta}} \ell(\mathbf{z}_l, y_l)) \|_2.
\end{aligned}
\tag{7}
$$

*Proof.* In this proof, we utilize one-step Newton approximation as an intermediary to estimate the error bound of the change in model parameters, *i.e.*,

$$
Err(-e_{ij}) = \left[ \mathcal{I}^*(-e_{ij}) - \mathcal{I}^{Nt}(-e_{ij}) \right] + \left[ \mathcal{I}^{Nt}(-e_{ij}) - \mathcal{I}(-e_{ij}) \right],
\tag{17}
$$

where $\mathcal{I}^*(-e_{ij}) = \Delta\hat{\theta}_\varepsilon = \hat{\theta}_\varepsilon - \hat{\theta}$, $\mathcal{I}^{Nt}(-e_{ij})$ is the one-step Newton approximation with the model parameter $\hat{\theta}_{Nt} = \hat{\theta} + \Delta\hat{\theta}_{Nt}$. According to Boyd et al. (2004) (Section 9.5.1), $\Delta\hat{\theta}_{Nt}$ can be calculated as follows:

$$
\begin{aligned}
\Delta\hat{\theta}_{Nt} = -\left(\mathbf{H}_{\hat{\theta}} + \lambda\mathbf{I}\right)^{-1} \cdot \frac{1}{N}\Big( & \sum_{i=1}^{N}\nabla_{\hat{\theta}}\ell(\mathbf{z}_i, y_i) + \sum_{v_l \in L}\nabla_{\hat{\theta}}\ell(\mathbf{z}'_l, y_l) \\
& - \sum_{v_l \in L}\nabla_{\hat{\theta}}\ell(\mathbf{z}_l, y_l) + \lambda\|\hat{\theta}\|_2\Big).
\end{aligned}
\tag{18}
$$

In the following, we will calculate the bound of $\mathcal{I}^*(-e_{ij}) - \mathcal{I}^{Nt}(-e_{ij})$ and $\mathcal{I}^{Nt}(-e_{ij}) - \mathcal{I}(-e_{ij})$ as two separate steps and combine them together. Here we define the before and after objective functions with the removal of edge $e_{ij}$ as follows:

$$
\begin{aligned}
\mathcal{L}_b(\theta) &= \sum_{i=1}^{n}\ell(\mathbf{z}_i, y_i) + \frac{\lambda}{2}\|\theta\|_2^2, \\
\mathcal{L}_a(\theta) &= \sum_{i=1}^{n}\ell(\mathbf{z}_i, y_i) + \sum_{v_l \in L}\ell(\mathbf{z}'_l, y_l) - \sum_{v_l \in L}\ell(\mathbf{z}_l, y_l) + \frac{\lambda}{2}\|\theta\|_2^2.
\end{aligned}
\tag{19}
$$

*Step I: Bound of $\mathcal{I}^*(-e_{ij}) - \mathcal{I}^{Nt}(-e_{ij})$.*

Due to that SGC model is convex on $\theta$, we take the second derivative of $\mathcal{L}_a(\theta)$ and have

$$
\lambda\mathbf{I} + \frac{1}{N}\left[\sum_{i=1}^{N}\nabla^2\mathcal{L}\left(\mathbf{z}_i, y_i\right) + \sum_{v_l \in L}\nabla^2\mathcal{L}\left(\mathbf{z}'_l, y_l\right) - \sum_{v_l \in L}\nabla^2\mathcal{L}\left(\mathbf{z}_l, y_l\right)\right] \succ 0.
\tag{20}
$$

To simplify the above equation, we define $\sigma'_{min}$ and $\sigma'_{max}$ are the smallest and largest eigenvalues of $\nabla^2\ell\left(\mathbf{z}'_l, y_l\right)$ and $\sigma_{min}$ and $\sigma_{max}$ are the smallest and largest eigenvalues of $\nabla^2\mathcal{L}\left(\mathbf{z}_l, y_l\right)$. Then we have

$$
\mathbf{I} \cdot \left(\lambda + \frac{(N - |L|) \cdot \sigma_{min} + |L| \cdot \sigma'_{min}}{N}\right) \succ 0.
\tag{21}
$$

Therefore, the SGC loss function corresponds to the removal of edge is strictly convex with the parameter $\left(\lambda + \frac{(N-|L|) \cdot \sigma_{min} + |L| \cdot \sigma'_{min}}{N}\right)$. By this convexity property and the implications of strong convexity (Boyd et al., 2004) (Section 9.1.2), we can bound $\mathcal{I}^*(-e_{ij}) - \mathcal{I}^{Nt}(-e_{ij})$ with the first derivative of SGC loss function as follows:

$$
\begin{aligned}
& \mathcal{I}^*(-e_{ij}) - \mathcal{I}^{Nt}(-e_{ij}) \\
= & \|\Delta\hat{\theta}_\varepsilon - \Delta\hat{\theta}_{Nt}\|_2 = \|(\Delta\hat{\theta}_\varepsilon + \hat{\theta}) - (\Delta\hat{\theta}_{Nt} + \hat{\theta})\|_2 = \|\hat{\theta}_\varepsilon - \hat{\theta}_{Nt}\|_2 \\
\leq & \frac{2N}{N\lambda + (N - |L|)\sigma_{min} + |L|\sigma'_{min}} \cdot \Big\| \frac{1}{N}\Big(\sum_{i=1}^{N}\nabla_{\hat{\theta}_{Nt}}\ell(\mathbf{z}_i, y_i) + \sum_{v_l \in L}\nabla_{\hat{\theta}_{Nt}}\ell(\mathbf{z}'_l, y_l) \\
& - \sum_{v_l \in L}\nabla_{\hat{\theta}_{Nt}}\ell(\mathbf{z}_l, y_l) + \lambda\|\hat{\theta}_{Nt}\|_2\Big) \Big\|_2.
\end{aligned}
\tag{22}
$$

If we take a close look at the second term in the above equation, we notice it is equal the first derivative of $\mathcal{L}_a(\theta)$, *i.e.*,

$$
\nabla_\theta\ell_a(\hat{\theta}_{Nt}) = \frac{1}{N}\Big(\sum_{k=1}^{N}\nabla_{\hat{\theta}_{Nt}}\ell(\mathbf{z}_k, y_k) + \sum_{v_l \in L}\nabla_{\hat{\theta}_{Nt}}\ell(\mathbf{z}'_l, y_l) - \sum_{v_l \in L}\nabla_{\hat{\theta}_{Nt}}\ell(\mathbf{z}_l, y_l) + \lambda\|\hat{\theta}_{Nt}\|_2\Big).
\tag{23}
$$

Therefore, we focus on bounding $\|\nabla_\theta\mathcal{L}_a(\hat{\theta}_{Nt})\|_2$ in the following.

$$
\begin{aligned}
\|\nabla_\theta\mathcal{L}_a(\hat{\theta}_{Nt})\|_2 &= \|\nabla_\theta\mathcal{L}_a(\hat{\theta} + \Delta\hat{\theta}_{Nt})\|_2 \\
&= \|\nabla_\theta\mathcal{L}_a(\hat{\theta} + \Delta\hat{\theta}_{Nt}) - \nabla_\theta\mathcal{L}_a(\hat{\theta}) + \nabla_\theta\mathcal{L}_a(\hat{\theta})\|_2 \\
&= \|\nabla_\theta\mathcal{L}_a(\hat{\theta} + \Delta\hat{\theta}_{Nt}) - \nabla_\theta\mathcal{L}_a(\hat{\theta}) - \nabla_\theta^2\mathcal{L}_a(\hat{\theta})\Delta\hat{\theta}_{Nt}\|_2
\end{aligned}
\tag{24}
$$

The above last equation holds due to the definition of $\Delta\hat{\theta}_{Nt}$ in Eq. (18).

For any continuous function $f$ and any inputs a and b, there exists $f(a+b) - f(a) - bf'(a) = \int_0^1 b \cdot (f'(a+bt) - f'(a))dt$. Based on that, we can rewrite $\|\nabla_\theta \mathcal{L}_a(\hat{\theta}_{Nt})\|_2$ as follows:

$$
\begin{aligned}
\|\nabla_\theta \mathcal{L}_a(\hat{\theta}_{Nt})\|_2 &= \|\nabla_\theta \mathcal{L}_a(\hat{\theta} + \Delta\hat{\theta}_{Nt}) - \nabla_\theta \mathcal{L}_a(\hat{\theta}) - \nabla_\theta^2 \mathcal{L}_a(\hat{\theta})\Delta\hat{\theta}_{Nt}\|_2 \\
&= \left\| \int_0^1 \Delta\hat{\theta}_{Nt}(\nabla_\theta^2 \mathcal{L}_a(\hat{\theta} + \Delta\hat{\theta}_{Nt} \cdot t) - \nabla_\theta^2 \mathcal{L}_a(\hat{\theta}))dt \right\|_2.
\end{aligned}
\tag{25}
$$

We assume the loss function $\ell$ on is twice differentiable and the second derivative of the loss function is Lipschitz continuous at $\theta$, with parameter $C$. Here C is controlled by the third derivative (Curvature) of the loss function $\ell$. Thus, we have

$$
\|\nabla_\theta^2 \ell(\theta_1) - \nabla_\theta^2 \ell(\theta_2)\|_2 \leq C \cdot \|\theta_1 - \theta_2\|_2.
\tag{26}
$$

Then we take Eq. (26) into Eq. (25) and have

$$
\begin{aligned}
delt&\|\nabla_\theta \mathcal{L}_a(\hat{\theta}_{Nt})\|_2 \\
&\leq \|NC\Delta\hat{\theta}_{Nt} \int_0^1 tdt\|_2 = \frac{NC}{2}\|\Delta\hat{\theta}_{Nt}\|_2^2 = \frac{NC}{2}\|\nabla_\theta^2 \mathcal{L}_a(\hat{\theta})^{-1} \cdot \nabla_\theta \mathcal{L}_a(\hat{\theta})\|_2^2 \\
&\leq \frac{NC}{2} \cdot \frac{N^2}{(N\lambda + (N - |L|) \cdot \sigma_{min} + |L| \cdot \sigma'_{min})^2} \cdot \|\sum_{l \in L}(\nabla_{\hat{\theta}}\ell(\mathbf{z}'_l, y_l) - \nabla_{\hat{\theta}}\ell(\mathbf{z}_l, y_l))\|_2^2.
\end{aligned}
\tag{27}
$$

The above last inequation holds according to the bound of $\nabla_\theta^2 \mathcal{L}_a(\hat{\theta})^{-1}$ and Eq. (19).

Combining Eq. (22), (23) and (27), we finish the bound of $\mathcal{I}^*(-e_{ij}) - \mathcal{I}^{Nt}(-e_{ij})$ as follows:

$$
\begin{aligned}
&\|\mathcal{I}^*(-e_{i,j}) - \mathcal{I}^{Nt}(-e_{i,j})\|_2 \\
&\leq \frac{N^3 C}{(N\lambda + (N - |L|)\sigma_{min} + \sigma'_{min}|L|)^3} \cdot \|\sum_{v_l \in L}(\nabla_{\hat{\theta}}\ell(\mathbf{z}'_l, y_l) - \nabla_{\hat{\theta}}\ell(\mathbf{z}_l, y_l))\|_2^2.
\end{aligned}
\tag{28}
$$

We finish Step I.

*Step II: Bound of $\mathcal{I}^{Nt}(-e_{ij}) - \mathcal{I}(-e_{ij})$.*

By the definition of $\mathcal{I}^{Nt}(-e_{ij})$ and $\mathcal{I}(-e_{ij}))$, we have:

$$
\begin{aligned}
&\mathcal{I}^{Nt}(-e_{ij}) - \mathcal{I}(-e_{ij}) \\
&= \left\{ \left( \lambda\mathbf{I} + \frac{1}{N}\left[ \sum_{k=1}^n \nabla_{\hat{\theta}}^2 \ell(\mathbf{z}_k, y_k) + \sum_{v_l \in L} \nabla_{\hat{\theta}}^2 \ell(\mathbf{z}'_l, y_l) - \sum_{v_l \in L} \nabla_{\hat{\theta}}^2 \ell(\mathbf{z}_l, y_l) \right] \right)^{-1} \right. \\
&\quad \left. - \left( \lambda\mathbf{I} + \frac{1}{N}\sum_{k=1}^n \nabla_{\hat{\theta}}^2 \ell(\mathbf{z}_k, y_k) \right)^{-1} \right\} \cdot \left\{ \sum_{v_l \in L}(\nabla_{\hat{\theta}}\ell(\mathbf{z}'_l, y_l) - \nabla\ell(\mathbf{z}_l, y_l)) \right\}.
\end{aligned}
\tag{29}
$$

For simplification, we use matrix $\mathbf{A}$, $\mathbf{B}$ and $\mathbf{C}$ for the following substitutions:

$$
\begin{aligned}
\mathbf{A} &= \lambda\mathbf{I} + \frac{1}{N}\left[ \sum_{k=1}^n \nabla_{\hat{\theta}}^2 \ell(\mathbf{z}_k, y_k) - \sum_{v_l \in L} \nabla_{\hat{\theta}}^2 \ell(\mathbf{z}_l, y_l) \right], \\
\mathbf{B} &= \frac{1}{N} \sum_{v_l \in L} \nabla_{\hat{\theta}}^2 \ell(\mathbf{z}'_l, y_l), \quad \text{and} \quad \mathbf{C} = \frac{1}{N} \sum_{v_l \in L} \nabla_{\hat{\theta}}^2 \ell(\mathbf{z}_l, y_l),
\end{aligned}
\tag{30}
$$

where $\mathbf{A}$, $\mathbf{B}$ and $\mathbf{C}$ are positive definite matrix and have the following properties:

$$
\begin{aligned}
\lambda + \frac{(N - |L|)\sigma_{max}}{N} &\succ \mathbf{A} \succ \lambda + \frac{(N - |L|)\sigma_{min}}{N}, \\
\frac{|L|\sigma'_{max}}{N} &\succ \mathbf{B} \succ \frac{|L|\sigma'_{min}}{N}, \quad \text{and} \quad \frac{|L|\sigma_{max}}{N} \succ \mathbf{C} \succ \frac{|L|\sigma_{min}}{N}.
\end{aligned}
\tag{31}
$$

Therefore, we have

$$\mathcal{I}^{Nt}(-e_{ij}) - \mathcal{I}(-e_{ij}) = ((\boldsymbol{A}+\boldsymbol{B})^{-1} - (\boldsymbol{A}+\boldsymbol{C})^{-1}) \cdot \left\{ \sum_{v_l \in L} (\nabla_{\hat{\theta}}\ell(\mathbf{z}_l', y_l) - \nabla_{\hat{\theta}}\ell(\mathbf{z}_l, y_l)) \right\}, \quad (32)$$

where $(\boldsymbol{A}+\boldsymbol{B})^{-1} - (\boldsymbol{A}+\boldsymbol{C})^{-1} \prec \frac{N}{N\lambda + (N-|L|)\sigma_{min} + |L|min(\sigma_{min}', \sigma_{min})}\boldsymbol{I}$.

The $l_2$ norm of the error between our predicted influence and Newton approximation can be bounded as follows:

$$\begin{aligned} &\|\mathcal{I}^{Nt}(-e_{ij}) - \mathcal{I}(-e_{ij})\|_2 \\ \leq &\frac{N}{N\lambda + (N-|L|)\sigma_{min} + \min(\sigma_{min}', \sigma_{min})|L|} \cdot \| \sum_{v_l \in L} (\nabla\ell_{\hat{\theta}}(\mathbf{z}_l', y_l) - \nabla\ell_{\hat{\theta}}(\mathbf{z}_l, y_l))\|_2. \end{aligned} \quad (33)$$

We finish Step II.

Combining the conclusion in Step I and II in Eq. (28) and (33), we have the error between the actual influence and our predicted influence as:

$$\begin{aligned} &Err(-e_{ij}) \\ \leq &\|\mathcal{I}^*(-e_{ij}) - \mathcal{I}^{Nt}(-e_{ij})\|_2 + \|\mathcal{I}^{Nt}(-e_{ij}) - \mathcal{I}(-e_{ij})\|_2 \\ = &\frac{N^3 C}{(N\lambda + (N-|L|)\sigma_{min} + |L|\sigma_{min}')^3} \cdot \| \sum_{v_l \in L} (\nabla_{\hat{\theta}}\ell(\mathbf{z}_l', y_l) - \nabla_{\hat{\theta}}\ell(\mathbf{z}_l, y_l))\|_2^2 \\ &+ \frac{N}{N\lambda + (N-|L|)\sigma_{min} + \min(\sigma_{min}', \sigma_{min})} \cdot \| \sum_{v_l \in L} (\nabla_{\hat{\theta}}\ell(\mathbf{z}_l', y_l) - \nabla_{\hat{\theta}}\ell(\mathbf{z}_l, y_l))\|_2. \end{aligned} \quad (34)$$

We finish the whole proof. □

**Corollary A.1.** *Let $\sigma_{min} \geq 0$ denote the smallest eigenvalue of all eigenvalues of Hessian matrices $\nabla_{\hat{\theta}}^2\ell(\mathbf{z}_i, y_i), \forall v_i \in V_{train}$ of the original model $\hat{\theta}$. Let $\sigma_{min}' \geq 0$ denote the smallest eigenvalue of all eigenvalues of Hessian matrices $\nabla_{\hat{\theta}(-v_i)}^2\ell(\mathbf{z}_i, y_i), \forall v_i \in V_{train}$ of the retrained model $\hat{\theta}(-v_i)$ with $v_i$ removed from graph $G$. Use $S$ denote the set $\{v : \mathbf{z}' \neq \mathbf{z}\}$ containing affected nodes from the node removal, and $Err(-v_i) = \|\mathcal{I}^*(-v_i) - \mathcal{I}(-v_i)\|_2$. We have the following upper bound on the estimated error of model parameters' change:*

$$\begin{aligned} Err(-v_i) \leq &\frac{N^3 m^2 C}{((N-1)\lambda + (N-|S|)\sigma_{min} + \sigma_{min}'|S|)^3} + \frac{(N-1)m}{N\lambda + (N-|S|)\sigma_{min} + \min(\sigma_{min}, \sigma_{min}')|S|} \\ &+ \frac{N^3 C}{(N\lambda + (N-1)\sigma_{min})^3} \cdot \|\ell(\mathbf{z}_i', y_i)\|_2^2 + \frac{N}{N\lambda + N\sigma_{min}} \cdot \|\ell(\mathbf{z}_i', y_i)\|_2 \end{aligned} \quad (35)$$

*where $m = \| \sum_{v_s \in S}[\nabla_{\hat{\theta}}\ell(\mathbf{z}_s', y_s) - \nabla_{\hat{\theta}}\ell(\mathbf{z}_s, y_s)] - \nabla_{\hat{\theta}}\ell(\mathbf{z}_i, y_i)\|_2$.*

*Proof.* We provide a simple proof for the error bound of removing a complete nodes. Notice that this error can be decomposed into two parts, 1, the error or removing a single node embedding $\mathbf{z}_i$ and 2, the error of adding $\mathbf{z}_s'$ and removing $\mathbf{z}_s$, where $s \in S$. where we have

$$Err(-v_i) \leq \sum_{s \in S} Err(z_s \to z_s') + Err(-z_i)$$

Notice that Eq. Theorem 4.1 proofs the error bound of $Err(-v_i)$, in the proving process we decompose the problem in to deriving the error bound by adding $\mathbf{z}_l'$ and removing $\mathbf{z}_l$ where $l \in L$, where $L$ is the set of changed node embedding caused by removing a edge from the graph. Following the same proving setting of Eq. Theorem 4.1, Again, notice that $S$ is the set of changed node embedding caused by removing a node from the graph. We simply substitute $L$ by $S$, we have the error bounds for $\sum_{s \in S} Err(z_s \to z_s')$.

$$\begin{aligned} \sum_{s \in S} Err(z_s \to z_s') \leq &\frac{N^3 m^2 C}{N\lambda + (N-|S|)\sigma_{min} + \sigma_{min}'|S|)^3} \\ &+ \frac{(N-1)m}{N\lambda + (N-|S|)\sigma_{min} + \min(\sigma_{min}, \sigma_{min}')|S|}, \end{aligned}$$

Where $m = \|\sum_{v_s \in S}[\nabla_{\hat{\theta}}\ell(\mathbf{z}'_s, y_s) - \nabla_{\hat{\theta}}\ell(\mathbf{z}_s, y_s)]\|_2$.

For $Err(-z_i)$, it can be derived following the same proving process as Eq. Theorem 4.1, but we only remove one data points. In this case we have:

$$Err(-z_i) \leq \frac{N^3 C}{(N\lambda + (N-1)\sigma_{\min})^3} \cdot \|\ell(\mathbf{z}'_i, y_i)\|_2^2 + \frac{N}{N\lambda + N\sigma_{min}} \cdot \|\ell(\mathbf{z}'_i, y_i)\|_2.$$

Combining the two error bounds we have:

$$Err(-v_i) \leq \frac{N^3 m^2 C}{((N-1)\lambda + (N-|S|)\sigma_{\min} + \sigma'_{\min}|S|)^3} + \frac{(N-1)m}{N\lambda + (N-|S|)\sigma_{\min} + \min(\sigma_{\min}, \sigma'_{\min})|S|}$$
$$+ \frac{N^3 C}{(N\lambda + (N-1)\sigma_{\min})^3} \cdot \|\ell(\mathbf{z}'_i, y_i)\|_2^2 + \frac{N}{N\lambda + N\sigma_{min}} \cdot \|\ell(\mathbf{z}'_i, y_i)\|_2$$

$$(36)$$

$\square$

## B  DATASET STATISTICS

We present the data statistic on our experiments below. We choose only small and medium-sized data. Because, each time we validate the influence of the elements in a graph, we need to retrain the model.

Table 4: Dataset Statistics

| Dataset | # Node | # Edge | # Class | # Feature | # Train/Val/Test |
|---|---|---|---|---|---|
| *Cora* | 2,708 | 5,429 | 7 | 1,433 | 140 / 500 / 1,000 |
| *Citeseer* | 3,327 | 4,732 | 6 | 3,703 | 120 / 500 / 1,000 |
| *Pubmed* | 19,717 | 44,338 | 3 | 500 | 60 / 500 / 1,000 |
| *WikiCS* | 11,701 | 216,123 | 10 | 300 | 250/ 1769 /5847 |
| *Amazon Computer* | 13,752 | 245,861 | 10 | 767 | 200 / 300 / Rest |
| *Amazon Photo* | 7,650 | 119,081 | 8 | 745 | 160 / 240 / Rest |

## C  VALIDATING INFLUENCE OF ELEMENTS: EXTRA DATASETS

For the *Wiki-CS* dataset, we randomly select one of the train/val/test split as described in Mernyei & Cangea (2020) to explore the effect of training nodes/edges influence. For the Amazon Computers and Amazon Photo dataset, we follow the implementation of Shchur et al. (2018). To set random splits, On each dataset, we use $20 * C$ nodes as training set, $30 * C$ nodes as validating set and the rest nodes as testing set, where $C$ is the number of classes.. Because for validating every edge's influence, we need to retrain the model and compare the change on loss, the computation cost is exceptionally high. We randomly choose 10000 edges of each datasets and validate their influence. We observe that even for medium-size datasets, our estimated influence is of high correlation to the actual influence.

## D  EMPIRICAL VERIFICATION OF THEOREM 4.1

As the value of $l_2$ regularization term decreases, the accuracy of our estimation of the influence of edges drops, and the Spearman correlation coefficient decrease correspondingly. This trend is consistent with the interpretations of the error bound on Theorem 4.1 that the estimation error of an influence function is inversely related with the $l_2$ regularization term. We also notice that the edges that connects high-degree nodes have overall less influence. Their estimation points lies relatively close to the $y=x$ line and thus could have relative small estimation error.

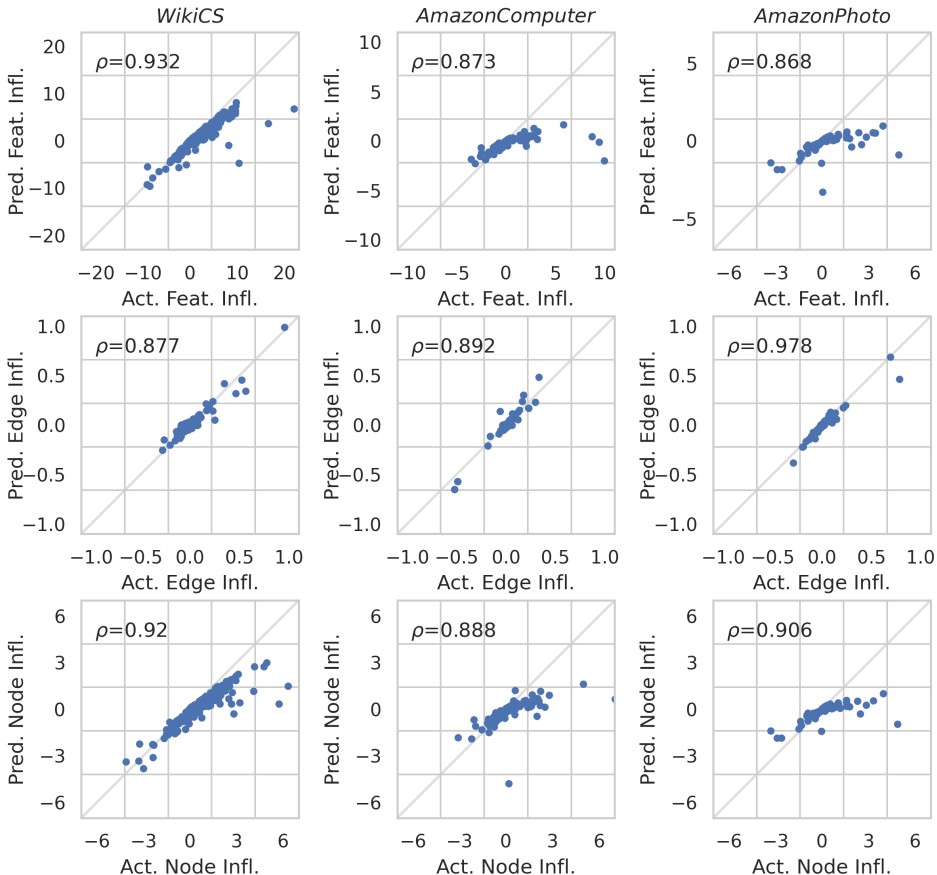

Figure 4: Estimated influence vs. actual influence on medium-sized graphs. Three datasets are used in this illustration *Wiki-CS* (left column), *Amazon Computers* (middle column) and *Amazon Photo* (right column). In all plots, the horizontal axes indicate the actual influence on the test set, the vertical axes indicate the predicted influence, and $\rho$ indicates Spearman's correlation coefficient between our predictions and the actual influences. **Top row**: Influence of node embeddings. **Middle row**: Influence of edge removals. Each point corresponds a removed training edge. **Bottom row**: Influence of node removal. Each point represents a removed training node.

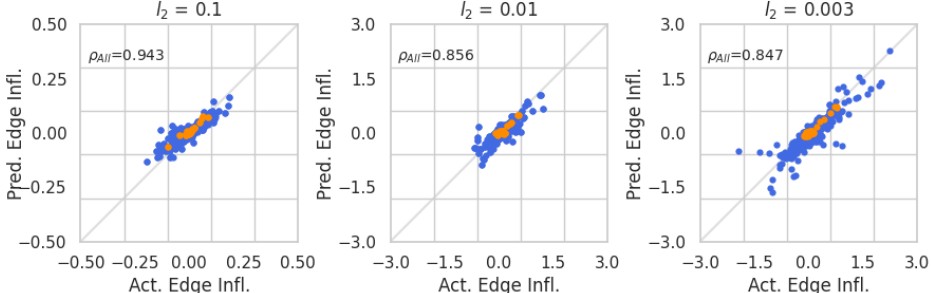

Figure 5: Spearman correlation on *Citeseer* dataset with different $l_2$ regularization term on validating influence of edges. The orange points denote the summations of the degrees of the two nodes that an edge connects is high. The blue points denote the edges, which are the summations of the degrees of the two nodes connecting the edge.

## E    GROUP EFFECT OF REMOVING MULTIPLE EDGES

We study the group effect of influence estimation on removing multiple edges. On dataset *Cora*, we randomly sample $k$ edges from the attributed graph, where $k$'s values were chosen increasingly as $2, 10, 50, 100, 200, 350$. Every time, we remove $k$ edges simultaneously and validate their estimated influence. We observe: though with high correlation, our influence plots tend to move downward as more edges are removed at the same time. In this case, our method tends to be less accurate and underestimates the influence of a simultaneously removed group of edges.

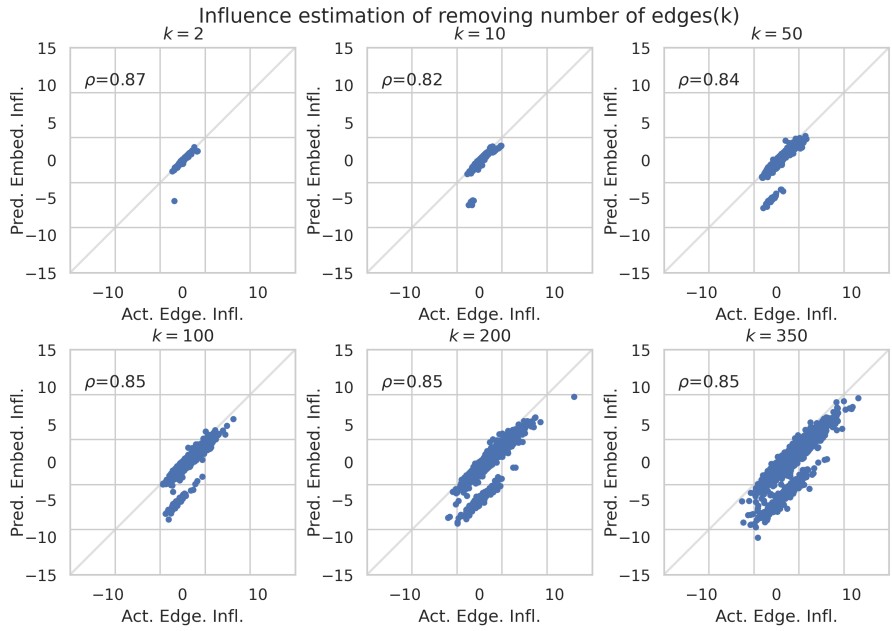

Figure 6: Estimating group influence on *Cora*. The horizontal axes indicate the actual influence on the validation set, and the vertical axes indicate the predicted influence. On each set, we randomly sample $k$ edges ($k$=2, 10, 50, 100, 200, 350) from the graph and repeat this process 5000 times. Each time, we remove k edges simultaneously and validate our influence estimation.

## F    VALIDATING INFLUENCE FOR ARTIFICIALLY ADDED EDGES

In this section, we validate our influence estimation for artificially added edges on dataset *Cora*, *Pubmed*, and *Citeseer*. On each dataset, we randomly select 10000 unconnected node pairs, add an artificial edge between them and validate its influence estimation. Figure 7 shows that the estimated influence correlates highly with the actual influence. This demonstrates that our proposed method can successfully evaluate the influence of artificially added edges.

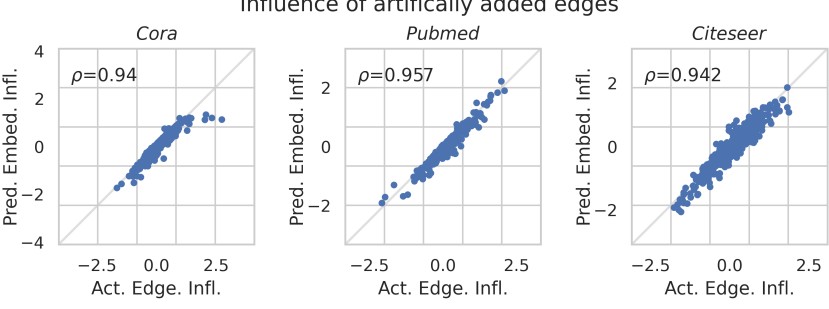

Figure 7: Estimated influence vs. actual influence on artificially added edges. Three datasets are used in this illustration *Cora*, *Pubmed*, and *Citeseer*. Due to the high time complexity of evaluating the influence on every pair of nodes, we randomly sample 10000 node pairs and add artificial edge.

## G   STUDY OF EDGES WITH NEGATIVE INFLUENCE

Here we demonstrate the performance via cumulatively removing edges with negative influence in Figure 8. The detailed implementation has been discussed in Section 5.3. Due to that the inaccurate influence estimation with more edges removed, we consider a maximum of 50 edges to be removed for each dataset. We observe an overall increase in model performance as we cumulatively remove edges predicted as a negative influence. This again demonstrates the usefulness of our influence estimation on edges.

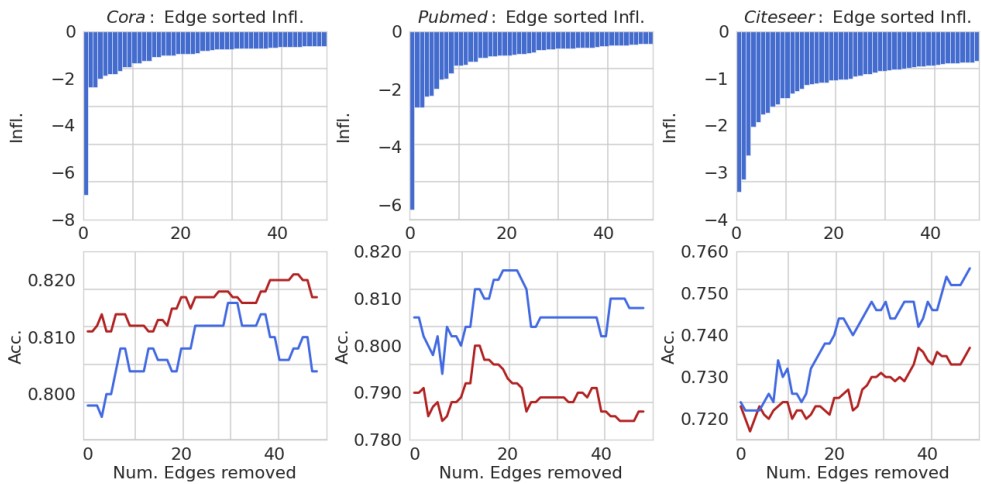

Figure 8: Study of edges with negative influence, each column corresponds to *Cora*, *Pubmed*, and *Citeseer* dataset. **Top**: the scale of edges with negative influence. **Bottom**: accuracy by cumulatively removing edges with negative influence. Blue and red lines present the accuracy changes of validation and test in response to negative influence edge removal, respectively.

## H   EXTEND INFLUENCE METHOD TO OTHER GNN MODELS

Theoretically, our current pipeline can be extended to other nonlinear GNNs under some violation of assumption. (1) According to Propositions Proposition 3.2 and Proposition 3.3, we require the existence of the inverse of the Hessian matrix, which is based on the assumption that the loss function on model parameters is strictly convex. Under the context of some GNN models with nonlinear activation functions, we can use the pseudo-inverse of the hessian matrix instead. (2) For non-convex loss functions of most GNN, our proposed error bound in Theorem Theorem 4.1 does not hold unless a large regularization term is applied to make the hessian matrix positive definite. From the implementation purpose, (1) From the implementation perspective, the non-linear models usually have more parameters than the linear ones, which require more space to store the Hessian matrix. Accordingly, the calculation of the inverse of the Hessian matrix might be out of memory. It needs to reformulate the gradient calculation and apply optimization methods like Conjugate gradient for approximation. (2) Our current pipeline is constructed based on mathematical, hands-on derived gradients adopted from Koh et al. (2019). Existing packages like PyTorch use automatic differentiation to get the gradients on model parameters. It could be inaccurate for second-order gradients calculation. Extending the current pipeline to other GNNs may require extensive first and second-order gradient formulations. We will explore more GNN influence in the future.

## I   RUNNING TIME COMPARISON

We present the running time comparison between calculating the edge influence via the influence-based method and retrieving the actual edge influence via retraining. We conduct our experiment on dataset *Cora*, *Pubmed*, and *Citeseer*. We demonstrate our method is 15-25 faster than the retrained method. Notably, for tasks like improving model performance or carrying out adversarial attacks via

edge removal, it could save a considerable amount of time in finding the edge to be removed with the lowest/largest influence.

Table 5: Running time comparisons for edge removal by second. Self-loop edges are not recorded.

| Dataset | Infl. (single edge) | Infl. (all edges) | Retrain (single edge) | Retrain (all edges) |
|---------|---------------------|-------------------|-----------------------|---------------------|
| *Cora* | $0.0049 \pm 0.0006$ | 24.86 | $0.0683 \pm 0.0216$ | 370.80 |
| *Pumbed* | $0.0008 \pm 0.0001$ | 34.58 | $0.0203 \pm 0.0044$ | 899.62 |
| *Citeseer* | $0.0097 \pm 0.0008$ | 45.90 | $0.1578 \pm 0.0404$ | 746.47 |

