# OpenReview forum: "Characterizing the Influence of Graph Elements"
_ICLR.cc/2023/Conference — ICLR 2023 poster_

### Official Review · Reviewer_Jtg1 · 2022-10-23

**Confidence:** 3
**Correctness:** 4
**Technical Novelty And Significance:** 2
**Empirical Novelty And Significance:** 3
**Recommendation:** 6

**Clarity, Quality, Novelty And Reproducibility:**

The work makes a tangible and well-presented contribution in the field of GCN training.

**Strength And Weaknesses:**

Strength:
- Novel theoretical results.
- Experimental verification over the retraining case.

Weakness:
- Limited to removals operations.
- Unclear and biased concept of rectification.

**Summary Of The Paper:**

This paper sets out to analyze the influence of attributed graph elements on the changes of parameters in a Graph Convolutional Network under Simple Graph Convolution, without having to go through retraining. The derived influence functions are capable to estimate changes caused by the removal of nodes or edges from the graph. A theoretical analysis provides bounds on the errors of such estimation.

**Summary Of The Review:**

This paper sets out to analyze the influence of attributed graph elements on the changes of parameters in a Graph Convolutional Network under Simple Graph Convolution, without having to go through retraining. The derived influence functions are capable to estimate changes caused by the removal of nodes or edges from the graph. A theoretical analysis provides bounds on the errors of such estimation.

It is not clear why the conducted analysis should be limited to removal operations.
Other operations, such as addition, or rewiring under some invariant, may also be studied to derive a complete framework.
Potential missing elements or predicted elements of a network are also candidate for such analysis.
In particular, it is not clear why the proposed concept of graph recitification should only involve removals.

---

> ### Author Response · Authors · 2022-11-16
> **Response to Reviewer Jtg1**
>
> We would like to thank Reviewer Jtg1 for your insightful review. Here we address your concerns in the following
>
> $\textbf{Limited to removals operations}$
>
> We completely agree with Reviewer Jtg1 that our pipeline can be enriched and more completed by considering more operations like adding edges or rewiring the graph. We did not involve them for the following reasons.
> - Adding an edge and removing an edge are essentially the same thing in terms of influence functions.
> - Although removing an edge takes roughly a similar time as adding an edge, the execution time of figuring out which edge to be removed or added towards a purpose is significantly different for a sparse graph since we need to check every possible edge to remove or add.
> - In some cases, adding an edge is inflexible or expensive, e.g., there is no chemical bond between two atoms.
>
> To address Reviewer Jtg1's concerns, we have added an extra experiment **Figure 7 in the appendix** to discuss the influence estimation of artificially added edges. It shows that the estimated influence correlates highly with the actual influence on artificially added edges. Therefore, we demonstrate our proposed method can successfully evaluate the influence of artificially added edges.

---

> > ### Comment · Reviewer_Jtg1 · 2022-11-18
> > **execution time difference**
> >
> > I guess you mean that the execution time to figure out which edge to remove is different from that to figure out which edge to add on a sparese graph. What impact does this difference have in terms of application?

---

> > > ### Author Response · Authors · 2022-11-18
> > > **Response to execution time**
> > >
> > > We are happy to see the response from Reviewer Jtg1 and feel more happier to conduct a discussion on this open problem.
> > >
> > > We come up with an application on a traffic graph, where we can close an existing road or construct a new one to alleviate congestion. Here we calculate the influence of an edge on the traffic congestion to seek which road to close or open. If we focus on closing one road, we need calculate $M$ times to traverse all the existing roads and find the one with the maximum influence. If we focus on opening a new road, we have to calculate all the possible new roads, i.e., $N^2-M$. Usually graphs are sparse,  $M\ll N^2-M$. Thus, we say the execution time of figuring out which edge to be removed or added towards a purpose is significantly different.
> > >
> > > Hope the above example makes our explanations more clearer. Thank you very much for your suggestions on other operations, which makes our paper more stronger!

---

### Official Review · Reviewer_pmjX · 2022-10-23

**Confidence:** 3
**Correctness:** 4
**Technical Novelty And Significance:** 3
**Empirical Novelty And Significance:** 2
**Recommendation:** 6

**Clarity, Quality, Novelty And Reproducibility:**

In terms of quality and clarity, the paper is in general well-written with minor typos. There are also sufficient experiments performed and an explanation of the setting used. Of course more should be done here: 1. Provide explanation on why some nodes or edges appear as outlines in Figure 4., or if there is a pattern for which edges/nodes are important (e.g. bridges/articulation points are they important?).
 In terms of originality, the paper is to the best of my knowledge the first to study influence functions, that became popular due to Koh & Liang, in GNNs. The fact that SGC is used as an exemplar GCN allows to re-use older results that do not involve the typical nonlinearities that exist in other GCN architectures. As a side note, results also assume additivity of incluence functions. I wonder the impact of this as there is dependency in graphs. E.g. if two edges connect two communities, based on the assumption: $\mathcal{I}(-x_i, -x_j) = \mathcal{I}(-x_i) + \mathcal{I}(-x_j)$ then potentially the impact of removing just one is minimal compared to removing both.

**Strength And Weaknesses:**

Strengths:
* The use of SGC allows the use of influence function in the context of GCNs.
* Despite its simplicity, it seems that results with respect to graph poisoning, can be transferred
to other architectures with non-linearities, like GCN from Kipf et al.
* Theoretical bounds on the error of approximation are also provided.

Weaknesses:
* The use of SGC essentially transforms the problem to that of deriving the influence function of a convex optimization problem.
Hence, what is important in this context is the graph structure itself and the smoothness parameter $k$. I didn't see them being involved in a clear and intuitive way in the paper. E.g. for which types of edges their influence function is higher/lower, or is it difficult to approximate? What is the effect of the smoothness parameter $k$ in the results?  It is difficult to see how Thm 4.1 provides any intuition to this direction.
* There is no mention of the running time (both actual and in asymptotic notation) of calculating the influence functions for each edge/node in the graph.
* There is a line of work, e.g. a paper from [NeurIPS 2021 ->" Robust Counterfactual Explanations on Graph Neural Networks"] or [KDD 21->"Counterfactual graphs from Explainable Classification of Brain Networks"], that identify a set of edges that removing them will change the prediction. Maybe the authors could compare with those in terms of indentifying influential edges (in the extreme positive/negative range).

**Summary Of The Paper:**

This paper proposes extending influence functions-- a method from statistics that approximates the change of parameters or loss functions, with respect to removing or modifying training instances -- for using them in GNNs. As typical GNNs usually involve non-linear activation functions that make derivations of formulas difficult, a simplified version of them, called SGC, is used. SGC is essentially a GNN stripped of all non-linear functions. Hence, the learnable weight matrices collapse to a single matrix that transforms a smoothed version of the adjacency matrix, A^k. For each edge and node, the authors calculate the influence of them (positive or negative) in the loss function. Based on this they propose two applications, graph rectification and graph poisoning, where their method can be useful.

**Summary Of The Review:**

I think this paper as it is falls within the threshold line. From one point the application of influence functions to GCNs is interesting with first promising results. On the other hand, though, little explanation is provided to the results (besides the accuracy of the method). Moreover, there is no mention at all on the running time and comparison with other methods that collect set of edges for counterfactual graph explanation.

---

> ### Author Response · Authors · 2022-11-16
> **Response to Reviewer pmjx**
>
> Thanks Reviewer pmjx for your insightful review :) Here we address your comments in the following.
>
> $\textbf{Smoothness parameter $k$}$
>
> First, we are a little bit confused about the smoothness parameter $k$ mentioned by the reviewer. If we understand correctly, $k$ means the $k$-hop of neighborhood nodes for message-passing. In this paper, we use 2-layer SGC for simplicity, which involves 2-hop neighbors for message-passing. When $k$ goes up, removing a single edge will affect a larger range of nodes. If this aligns with what Reviewer pmjx thoughts, we would be happy to add a corollary on this point.
>
> ---
>
> $\textbf{Types of edges their influence function is higher/lower}$
>
> We provide further interpretations of the proposed error bound in the context of graph structures in Appendix D (Empirical verification of Theorem 4.1). We empirically found the edges with a relatively large degree may have relatively small influences. Specifically, we calculate the degree of each edge in the attributed graph by summing the degree of their connected nodes. Then, we compare the estimation error between the top 10% largest degree of edges (colored as orange) and the rest edges(colored as blue). We notice that the edges that connect high-degree nodes have less influence overall, and their Pred. vs. Act. Estimation points lie relatively close to the y=x line and thus could have a relatively small estimation error. Our interpretations of Remark 4.2 partially explain the observation. The corresponding change in node embedding should be relatively small when removing the edge with high degrees. Thus, its gradients change should also be small. Therefore, it enjoys a small estimation error.
>
> In addition, we estimate the edge influence and compare it to the actual influence under different values of $l_2$ regularization term (this may address the reviewer's concern if the smoothness parameter refers to the regularization term). We notice that as the $l_2$ value exponentially decreases, the Pred. vs. Act. estimation points tend to spread out is inversely related to the $l_2$ values.
>
> ---
>
> $\textbf{Running Time}$
>
> We appreciate the kind reminders from Reviewer pmjx to mention the running time. We have added an extra discussion on the running time comparison in **Appendix I**. We present the running time comparison between calculating the predicted edge influence via our proposed influence-based method and retrieving the actual edge influence via retraining on the $\textit{Cora}$, $\textit{Pubmed}$, and $\textit{Citeseer}$. When trying each specific candidate edge, our method is 15-25 faster than the retrained methods. Notably, for tasks like improving model performance or carrying out adversarial attacks via cumulatively removing edges, the trained methods need to go through every edge on each removal. However, our method only needs to estimate each edge's influence values by calculating its influence score and hence will be significantly faster than the retrained methods. The rate of speed-up should be in the order of the edge number in the graph and depends on the tasks.
>
> ---
>
> $\textbf{Two key references}$
>
> Thank you for providing these two key references. We would like to bring them into our manuscript for a detailed discussion. These two papers are located in the field of graph interpretation, which seeks the counterfactual graph/subgraph to provide explanations for the predictive results. In their papers, the counterfactual is defined as a graph that, while having high structural similarity with the original graph, is classified by the black box in a different class. On the contrary, our paper addresses quantifying the positive/negative influence of a single edge or node on a learning task. From the research problems, these two papers focus on the graph classification that trains their model on a set of graphs, while ours target the node classification where there is only one graph during the training. Therefore, it is not straightforward for us to align their algorithms into our settings or align our algorithm to their settings for fair comparisons. We would be quite appreciative if Reviewer pmjx is willing to shed some hints on this. We can quickly run the experiment and report the results during the response period or in the final version, following their public codes. Though we could not specifically discuss them for page limitations at this point, we enriched the paragraph 'understanding graph data' by adding the above papers as a reference.
>
> ---

---

> > ### Author Response · Authors · 2022-11-16
> > **Continue to the response to Reviewer pmjx**
> >
> >
> > $\textbf{Additivity}$
> >
> > Actually, we do not hold the additivity assumption on the influence of removing multiple edges, i.e., $\mathcal{I}(-e_{ij}-e_{i'j'}) =\mathcal{I}(-e_{ij})+\mathcal{I}(-e_{i'j'})$. Instead, our influence estimation is constructed under the additivity assumption of node embeddings. Removing multiple edges brings in an extra group effect [2]. To provide more understanding of the group effect for graph data, we have added an extra experiment to validate the group effects of removing several edges at the same time. We invited Reviewer pmjx to see **the newly added Figure 6 in the appendix**. As expected, we notice that our influence estimation tends to be less accurate with an increase in the number of removed multiple edges.
> >
> > ---
> >
> > [1] Koh, P. W., \& Liang, P. (2017, July). Understanding black-box predictions via influence functions. In International conference on machine learning (pp. 1885-1894). PMLR.
> >
> > [2] Koh, P. W. W., Ang, K. S., Teo, H., \& Liang, P. S. (2019). On the accuracy of influence functions for measuring group effects. Advances in neural information processing systems, 32.

---

> > > ### Comment · Reviewer_pmjX · 2022-11-28
> > > **Thanks for the detailed response**
> > >
> > > I would like to thank the authors for their detailed response. Regarding $k$, indeed I refer to the k-hop neighborhood used. The reason is that, as has been observed in past works (1), as long as $\tilde{S}$ has operator norm <= 1, then as $k \to \infty$
> > > , then each column of $S^{k}X$ converges to the dominant eigenspace of S, thus it is independent of the feature matrix X. In that case, one would typicalaly expect that the outcome is more susceptible to edge addition/removal for example, as you mention. I was curious for which values of k is it reasonble to expect good performance of the proposed method. Finally, I appreciate the time spent to address other concerns regarding the empirical evaluation and interpretability on the results. As my initial recommendation was already leaning positive towards the paper and certain concerns regarding the empirical evaluation and the lack of certain baselines remain, I will keep my score for the moment.
> > >
> > > (1)J.Jia & A.Benson: A Unifying Generative Model for Graph Learning Algorithms: Label Propagation, Graph Convolutions, and Combinations

---

> > > > ### Author Response · Authors · 2022-11-30
> > > > **Further response to Reviewer pmjx**
> > > >
> > > > We are happy to receive Reviewer pmjx's insightful response and appreciate the clarification on the smoothness parameter $k$. We completely agree with Reviewer pmjx's that as the number of $k$ increases to infinity, $S^{k}X$ may converge to the dominant eigenspace of $S$, which results in the over-smoothing issue in GNNs. Having this issue, all nodes may end up with similar node embeddings, and the performance of most GNNs may decrease accordingly.
> > > >
> > > > ---
> > > >
> > > > Respectfully, we would first kindly point out that the primary purpose of our work is to estimate the changes in model parameters and certain functions regarding model parameters (like the loss) as the results of removing elements (mathematically, $S$$\rightarrow$$S_{-e_{ij}}$ or $S$$\rightarrow$$S_{v}$) in graphs like nodes or edges without retraining the model. Thus, it is different from the GNN learning problem in the first paragraph, i.e., the drop in GNNs' performance regarding over-smoothing is kind of out of the scope of this discussion. Here, the performance of our method refers to how accurate our estimation of the change in validation/test loss is.
> > > >
> > > > ---
> > > >
> > > > To follow Reviewer pmjx's suggestion, we further study the impact of $k$ on our estimation task, where we draw Predicted Influence vs. Actual Influence to validate the correctness of our estimation (See Figure 2) and use the Spearman Correlation Coefficient to measure the performance of our estimation. Intuitively, as the number of $k$ gets large, removing a single edge will result in a change in a larger number of nodes. This, however, may result in a drop in our estimation accuracy. We add an extra experiment to discuss how the Spearman Correlation Coefficient changes in response to the increase of the smooth parameter $k$ ($k$$=$$2, 3, 4$) on dataset $Cora$ in the setting of edge removal. Also, we report the average number of changed training nodes in response to a single edge removal. We can see $k$$=$$2$ enjoys the best performance. For further experiments like improving model performance and carrying out adversarial attacks, we choose $k$$=$$2$ to demonstrate the effectiveness of our methods.
> > > >
> > > > | Spearman Coefficient of the estimation and retrain on the Cora dataset  | 2-layer | 3-layer | 4-layer |
> > > > |-------------------------------------------------------------------------|---------|---------|---------|
> > > > | Spearman Coef.                                                          | 0.88    | 0.58    | 0.56    |
> > > > | Ave. Num. Changed in training node                                      | 4.5     | 135.0   | 135.0   |
> > > >
> > > > ---
> > > >
> > > > Again, we would sincerely thank Reviewer pmjx's insightful comments and hope the above responses address your concerns.

---

> > > > > ### Comment · Area_Chair_2Z7k · 2022-12-01
> > > > > **To reviewer pmjX**
> > > > >
> > > > > Dear Reviewer. Authors,
> > > > >
> > > > > Thanks for the great in-depth discussion. Reviewer pmjX, does the above answer your concerns? If not, please describe what you feel is missing.
> > > > >
> > > > > Thanks,
> > > > > AC

---

> > > > > > ### Comment · Reviewer_pmjX · 2022-12-08
> > > > > > **Thanks for the reply**
> > > > > >
> > > > > > Yes, this adresses my comment. My point was that the impact of $k$ to the performance of the method should be mentioned, not necessarily a s a weakness of the current work, rather than keeping readers aware and a motivation for further studies.

---

### Official Review · Reviewer_9bmE · 2022-10-24

**Confidence:** 4
**Correctness:** 4
**Technical Novelty And Significance:** 3
**Empirical Novelty And Significance:** 3
**Recommendation:** 8

**Clarity, Quality, Novelty And Reproducibility:**

Quality: I think this is a high-quality paper. The authors address an important, challenging open question in the GNN literature. And they do so by adapting a well-founded method.
Clarity: This is one of the clearest and focused papers I have reviewed this year (out of 16 papers or so). It is clear what problem the authors are trying to solve and how they plan to solve it. Many papers I have read recently try to do too much and go in too many differen directions, and the main idea gets lost.
Originality: The application of influence functions for this task is novel to the best of my knowledge.

**Strength And Weaknesses:**

Strengths:
- Novelty. Interpretability of graph neural network models is an open area of research, and, to the best of my knowledge, influence functions have not been explored before for this task.
- Rigorous theoretical bounds on estimation error, and reasonable empirical results that show that theoretical bounds are relevant in practice.
- Practical significance. The authors illustrate that their proposed method can be used against / for adversarial attacks.
- Paper is very well-organized, and the authors stay focused on the task at hand.


Weaknesses:
- Gap between theory and practice. The paper focuses on SGC, presumably because it is easier to derive theoretical results for this model. SGC is not as popular as other graph architectures, such as GCN and its many variants. It is also not clear how the proposed method would work in the presence of skip connections, different normalizations, such as LayerNorm, or the myriad of enhancements that are commonly used in modern graph neural network models. As such, there is a gap between the methods proposed in the paper and application. However, this is to be expected of a first exploration of influence functions for GNNs.

**Summary Of The Paper:**

The authors propose a method to quantify and estimate how various components of graph neural network models will be affected by node/edge removals in the training set. In particular, the authors derive influence functions for the Simple Graph Convolution (SGC) architecture. The authors provide rigorous theoretical results, including error bounds on the estimated change vs. the actual effect of node/edge removals. A case study on adversarial attacks on graph neural networks showcase a very relevant application of the proposed method.

**Summary Of The Review:**

Please see the two sections above. Based on those sections, I believe this paper should be accepted.

---

> ### Author Response · Authors · 2022-11-16
> **Response to Reviewer 9bme**
>
> We would like to thank Review 9bmE's kind acknowledgment and affirmation. As the reviewers stated, our paper derives the influence of graph elements based on SGC, which is expected to be the first exploration of influence functions for GNNs. Despite the current implementations, our pipeline can be extended in the context of other GNNs' model architecture by relaxing some assumptions.
>
> - According to Propositions 3.2 and 3.3, we require the existence of the inverse of the Hessian matrix, which is based on the assumption that the loss function on model parameters is strictly convex. Under the context of some GNN models with non-linear activation functions, we can use the pseudo-inverse of the hessian matrix instead.
>
> - From the implementation perspective, the non-linear models usually have more parameters than the linear ones, which require more space to store the Hessian matrix. Accordingly, the calculation of the inverse of the Hessian matrix might encounter some space issues. It needs to further reformulate the gradient calculation and apply optimization techniques like Conjugate gradient for approximation.
>
> We agree the mentioned myriad of enhancements, like skip connections and normalizations, are important to GNNs. From our point of view, there is no conspicuous barrier to integrating these architectural improvements into the extended GNNs framework elaborated above as long as we can access the inverse hessian matrix or compute the gradients. We believe the reviewer's concerns can be successfully addressed by the new versions of influence estimation of our future work.

---

### Official Review · Reviewer_Lyjj · 2022-10-25

**Confidence:** 4
**Correctness:** 3
**Technical Novelty And Significance:** 3
**Empirical Novelty And Significance:** 3
**Recommendation:** 6

**Clarity, Quality, Novelty And Reproducibility:**

Overall, the paper is well-written, and the different concepts are clearly presented. At the same time, the paper keeps a good balance between theoretical contributions and empirical evaluation.  Despite building on existing ideas, the paper studies a novel application of influence functions in GNNs.

Typos:
* Page 4: Proposition 3.2 offer*s
* Section 5.4: Table 3 is not mentioned in the paragraphs of the subsection.


**Strength And Weaknesses:**

*Strengths:*
* Leveraging influence functions to analyze the performance of a GNN constitutes an interesting theoretical framework.

* The paper provides a good experimental pipeline in which the influence scores of nodes and edges are utilized in different ways (e.g., to define attacks).

*Weaknesses:*
* The formulation of the methodology is done on the Simplified Graph Convolution (SGC) model. Although the paper states that their analysis could be extended to other GNN models, this does not seem to be straightforward.

* Missing possible connections to related work and theory. One of the practical applications of influence functions considered in the paper, examines how edge removals could improve classification performance. To do so, edges with negative influence are removed from the graph before training the SGC model.

  First, what is the fraction of edges that are being removed, and how is the performance affected if we increase/decrease the number of deleted edges?

  Second, how is this methodology related to other approaches that remove edges towards building more robust models? For instance, it has been shown that a simple random edge removal could improve performance by reducing over-smoothing (the DropEdge model). Is this something contradictory to the theoretical results presented in the paper? This point needs further clarification to understand the impact of edge influence scores.


**Summary Of The Paper:**

The paper proposes a methodology to leverage influence functions toward estimating how GNN model parameters change under graph perturbation schemes. This is an important problem to efficiently understand the effect of graph elements on model parameters, without the need to retrain the model. Specifically, the paper considers edge and node removals as perturbation strategies, deriving theoretical bounds to characterize the changes on model parameters. Besides, it is empirically shown how to utilize such graph influence functions to (i) improve the prediction performance, and (ii) as tools to guide adversarial attacks to the GNN.

**Summary Of The Review:**

Overall, the paper introduces an exciting approach to further understanding the impact of graph components on the performance of the model. Nevertheless, I still have some concerns about the practical application of the framework (restricted or not to SGC) and its relationship to previous developments (e.g., DropEdge).

---

> ### Author Response · Authors · 2022-11-16
> **Response to Reviewer Lyjj**
>
> We would like to thank Reviewer Lyjj for your insightful review. Here we address your concerns in the following.
>
> ---
>
> $\textbf{Extend to other GNN models}$
>
> Our pipeline can be extended to non-linear GNN under several aspects' mild violation of assumptions.
>
> - According to Propositions 3.2 and 3.3, we require the existence of the inverse Hessian matrix, which is from the assumption that the loss function on model parameters is strictly convex. Under the context of some GNN models with non-linear activation functions, we can alternatively use the pseudo-inverse of the hessian matrix instead.
>
> - In the implementation, the non-linear models usually have more parameters than the linear ones, which require space to up $O(N^2)$ to store the Hessian matrix, where $N$ is the number of model parameters. For a huge model, there could be some space issues. It needs to reformulate the gradient calculation and apply optimization methods like Conjugate gradient for approximation. Inevitably, the above two points may lead to loose influence estimations. We have added discussions on extending to other GNN models in Appendix section H and will explore more in the future.
>
> ---
>
> $\textbf{Fraction of edges that are being removed}$
>
> In experiments, the numbers of removed edges on $\textit{Cora}$, $\textit{Pubmed}$, and $\textit{Citeseer}$ are 30, 19, and 50, respectively. The fractions are 0.55%, 0.04%, and 1.05%, respectively. We shall add these details to the experiment section.
>
> As for some further clarifications, we do not specify the number of edges that are being removed. Instead, we set this number according to the validation set. Specifically, we select the number of removed edges according to the highest accuracy on the validation set and report the corresponding accuracy on the test set (See lines 10-13 of the second paragraph in Section 5.3).
>
> Moreover, similar to Figure 3 (Study of edges with $\textit{positive}$ influence on both validation and test set), we have provided **extra experiments on edge removal in Figure 8 in the appendix** (Study of edges with $\textit{negative}$ influence on both validation and test set). Figure 8 shows an overall increase in model performance as we remove edges with negative influence cumulatively. In contrast to Figure 3, where the model performance can be decreased by removing positive influence edges, we again demonstrate the effectiveness of our influence estimation on graph elements.
>
> ---
>
> $\textbf{Relations to DropEdge}$
>
> Thank you for bringing DropEdge [1] to our attention! This paper is indeed a very interesting solution for edge removal. Our work is essentially different from DropEdge in the following aspects.
> - Purpose. As you mentioned, the significant contribution of the DropEdge is that it could reduce the over-smoothing issues and increase the model generalization. In our paper, we directly focus on removing edges that bring in negative effects on model performance.
>
> - The way of removing edges. DropEdge randomly removes some edges while we conduct edge removal according to their influence values. Our approach is more pertinence for selecting the edges to be removed.
>
> - Fraction of edges that are being removed. From the above response, the fractions of edges that are being removed are almost less than 1%, while the fractions of DropEdge vary from 20%-80%.
>
> The above statement is not contradicting the DropEdge model because we selectively remove edges instead of randomly removing them with different purposes. Instead, our influence-based pipeline can guide the DropEdge model by avoiding removing the edges identified as a positive influence.
>
> ---
>
> $\textbf{Typos}$
>
> All fixed. Thanks for your careful reading!
>
> ---
>
> [1] Rong, Y., Huang, W., Xu, T., \& Huang, J. (2019). Dropedge: Towards deep graph convolutional networks on node classification. arXiv preprint arXiv:1907.10903.

---

### Decision · Program_Chairs · 2023-01-20

**Decision:**

Accept: poster

**Justification For Why Not Higher Score:**

The scope of the work is somewhat limited, since understanding the effect of adversarial removal of edges and nodes may not be of broad interest to the community.

**Justification For Why Not Lower Score:**

All reviewers agree the work makes important contributions.

**Metareview: Summary, Strengths And Weaknesses:**

The work proposes a method to use influence functions to estimate how GNN model parameters change when the underlying training graph is perturbed by node/edge removals in the training set. This is an important task, where the authors derive theoretical bounds to characterize the changes on model parameters. A compelling empirical validation is used on adversarial attacks on graph neural networks to validate the proposed method.

All reviewers agree that the task is important and the contributions of this work are solid.



**Note From Pc:**

if the above contains the word "oral" or "spotlight" please see: "oral" presentation means -> notable-top-5% and "spotlight" means -> notable-top-25%. As stated in our emails, we are disassociating presentation type from AC recommendations